# EED orchestration of heart maturation through interaction with HDACs is H3K27me3-independent

**Shanshan Ai[1†], Yong Peng[1†], Chen Li[1], Fei Gu[2], Xianhong Yu[1], Yanzhu Yue[1], Qing Ma[2], Jinghai Chen[2], Zhiqiang Lin[2], Pingzhu Zhou[2], Huafeng Xie[3,7], Terence W Prendiville[2§], Wen Zheng[1], Yuli Liu[1], Stuart H Orkin[3,4,7,5], Da-Zhi Wang[2,4], Jia Yu[6], William T Pu[2,4*‡], Aibin He[1*‡]**

[1]Institute of Molecular Medicine, Peking-Tsinghua Center for Life Sciences, Beijing Key Laboratory of Cardiometabolic Molecular Medicine, Peking University, Beijing, China; [2]Department of Cardiology, Boston Children's Hospital, Boston, United States; [3]Division of Hematology/Oncology, Boston Children's Hospital, Boston, United States; [4]Harvard Stem Cell Institute, Harvard University, Cambridge, United States; [5]Howard Hughes Medical Institute, Boston, United States; [6]Department of Biochemistry and Molecular Biology, State Key Laboratory of Medical Molecular Biology, Institute of Basic Medical Sciences Chinese Academy of Medical Sciences, Peking Union Medical College, Beijing, China; [7]Department of Pediatric Oncology, Dana-Farber Cancer Institute, Boston, United States

*For correspondence: wpu@pulab.org (WTP); ahe@pku.edu.cn (AH)

[†]These authors contributed equally to this work
[‡]These authors also contributed equally to this work

Present address: [§]Department of Paediatric Cardiology, Our Lady's Children's Hospital Crumlin, Dublin, Ireland

Competing interests: The authors declare that no competing interests exist.

**Abstract** In proliferating cells, where most Polycomb repressive complex 2 (PRC2) studies have been performed, gene repression is associated with PRC2 trimethylation of H3K27 (H3K27me3). However, it is uncertain whether PRC2 writing of H3K27me3 is mechanistically required for gene silencing. Here, we studied PRC2 function in postnatal mouse cardiomyocytes, where the paucity of cell division obviates bulk H3K27me3 rewriting after each cell cycle. EED (embryonic ectoderm development) inactivation in the postnatal heart (Eed[CKO]) caused lethal dilated cardiomyopathy. Surprisingly, gene upregulation in Eed[CKO] was not coupled with loss of H3K27me3. Rather, the activating histone mark H3K27ac increased. EED interacted with histone deacetylases (HDACs) and enhanced their catalytic activity. HDAC overexpression normalized Eed[CKO] heart function and expression of derepressed genes. Our results uncovered a non-canonical, H3K27me3-independent EED repressive mechanism that is essential for normal heart function. Our results further illustrate that organ dysfunction due to epigenetic dysregulation can be corrected by epigenetic rewiring.

## Introduction

Normal developmental maturation of organ function requires precise transcriptional regulation of gene expression. This transcriptional regulation depends upon the interplay of an array of epigenetic regulators, which shape the epigenetic landscape by repositioning nucleosomes and depositing covalent modifications on histones. Since the chromatin landscape is established through a series of steps, each contingent upon normal completion of prior steps, transient disruption, through environmental mishaps or genetic mutations, might be anticipated to break the normal sequence and irreversibly impact organ development and function. Whether or not this is the case is presently unknown, and the answer has clear therapeutic implications for diseases that involve epigenetic changes.

Development and function of the heart is vulnerable to epigenetic insults, as mutation of epigenetic regulators causes both structural heart disease and cardiomyopathy in humans and in experimental model systems (*He et al., 2012a*; *Zaidi et al., 2013*; *Hang et al., 2010*; *Delgado-Olguín et al., 2012*; *Lickert et al., 2004*; *Montgomery et al., 2007*). Recent work has highlighted the critical role of epigenetic silencing of ectopic transcriptional programs in normal heart development and function (*He et al., 2012a*; *Delgado-Olguín et al., 2012*; *Montgomery et al., 2007*; *Trivedi et al., 2010*). One class of epigenetic repressors are the histone deacetylases (HDACs), which remove activating histone acetylation marks to repress gene expression. HDACs consist of four classes (classes I, IIa, IIb, and IV) on the basis of their domain structure and expression pattern (*Laugesen and Helin, 2014*; *Chang and Bruneau, 2012*). Zhao and colleagues reported that HDACs target both actively transcribed and repressed genes to reset chromatin state for subsequent complex-dependent transcriptional outcome (*Wang et al., 2009*). Consistent with this, class I HDACs such as HDAC1 and HDAC2 are frequently subunits of multi-protein complexes, such as Sin3, NuRD and CoREST. Inactivation of HDAC1/2 caused abnormal heart growth and function that was linked to ectopic expression of slow twitch skeletal muscle genes (*Montgomery et al., 2007*). Class II HDACs were found to repress cardiac MEF2 transcription factor activity and their genetic inactivation caused pathological cardiac hypertrophy (*Zhang et al., 2002*).

Another class of epigenetic repressors is polycomb repressive complex 2 (PRC2), comprising the core subunits EED, SUZ12, and either EZH1 or EZH2. Canonically, PRC2 represses gene transcription by catalyzing trimethylation of histone H3 on lysine 27 (H3K27me3) (*Cao et al., 2002*). However, whether or not H3K27me3 deposition is sufficient to account for PRC2-mediated transcriptional repression in all contexts remains uncertain. Margueron and colleagues reported that elevated EZH2 and H3K27me3 levels are a consequence of high proliferation of cells in tumorigenesis, arguing that perturbation of this equilibrium leads to irreversible transcriptional change (*Wassef et al., 2015*). In PRC2-null ESCs as well as other cell types, most genes that are marked by H3K27me3 remain repressed even after inactivation of PRC2 components, suggesting redundant repressive mechanisms. Furthermore, most studies of PRC2 function have been performed in actively cycling cells, whereas the majority of cells in adult mammals cycle slowly, and some cells, such as adult cardiomyocytes (CMs), are largely post-mitotic. Since cell cycle activity mandates re-deposition of histone marks (*Margueron et al., 2009*; *Hansen et al., 2008*), in mitotic cells writing activity of epigenetic complexes may overshadow other functions that are important in non-proliferating cells.

The role of PRC2 in the heart has been studied by cardiac-specific inactivation of EZH2 or EED during early cardiac development. This resulted in disruption of H3K27me3 deposition, derepression of non-cardiomyocyte gene programs, and lethal heart malformations (*He et al., 2012a*; *Delgado-Olguín et al., 2012*). Here, we investigated the function of EED in postnatal heart maturation. Postnatal CMs have largely exited the cell cycle (*Bergmann et al., 2009*, *Bergmann et al., 2015*; *Senyo et al., 2013*; *Ali et al., 2014*; *Alkass et al., 2015*; *Soonpaa et al., 2015*; *Hirai et al., 2016*) and express little EZH2 protein (*He et al., 2012a*), but nonetheless contain abundant H3K27me3. Thus, the postnatal heart affords a unique opportunity to investigate the mechanism by which EED represses gene expression in post-mitotic cells and to harness this knowledge to determine whether the dysregulated chromatin landscape can be rewired to restore gene expression and heart function. We found that EED silences the slow-twitch myofiber gene program to orchestrate heart maturation by complexing with and stimulating HDAC deacetylase activity. Surprising, de-repression of the majority of genes in EED-deficient cardiomyocytes occurred without loss of H3K27me3, pointing to important EED repressive activity that is independent of its role in PRC2-mediated H3K27 trimethylation.

## Results

### Heart dysfunction in neonatal cardiac-inactivation of Eed

To understand the role of *Eed* in regulating cardiac gene expression during heart maturation, we generated *Eed*fl/fl; *Myh6*Cre (EedCKO) mice, in which *Myh6*Cre specifically inactivates the conditional *Eed*fl allele in CMs. Western blotting and qRTPCR demonstrated effective cardiac EED protein depletion that occurred between postnatal day (P) 0 and 5 (*Figure 1A* and *Figure 1—figure supplement 1*). EedCKO mice were born at the expected Mendelian frequency, suggesting a lack of

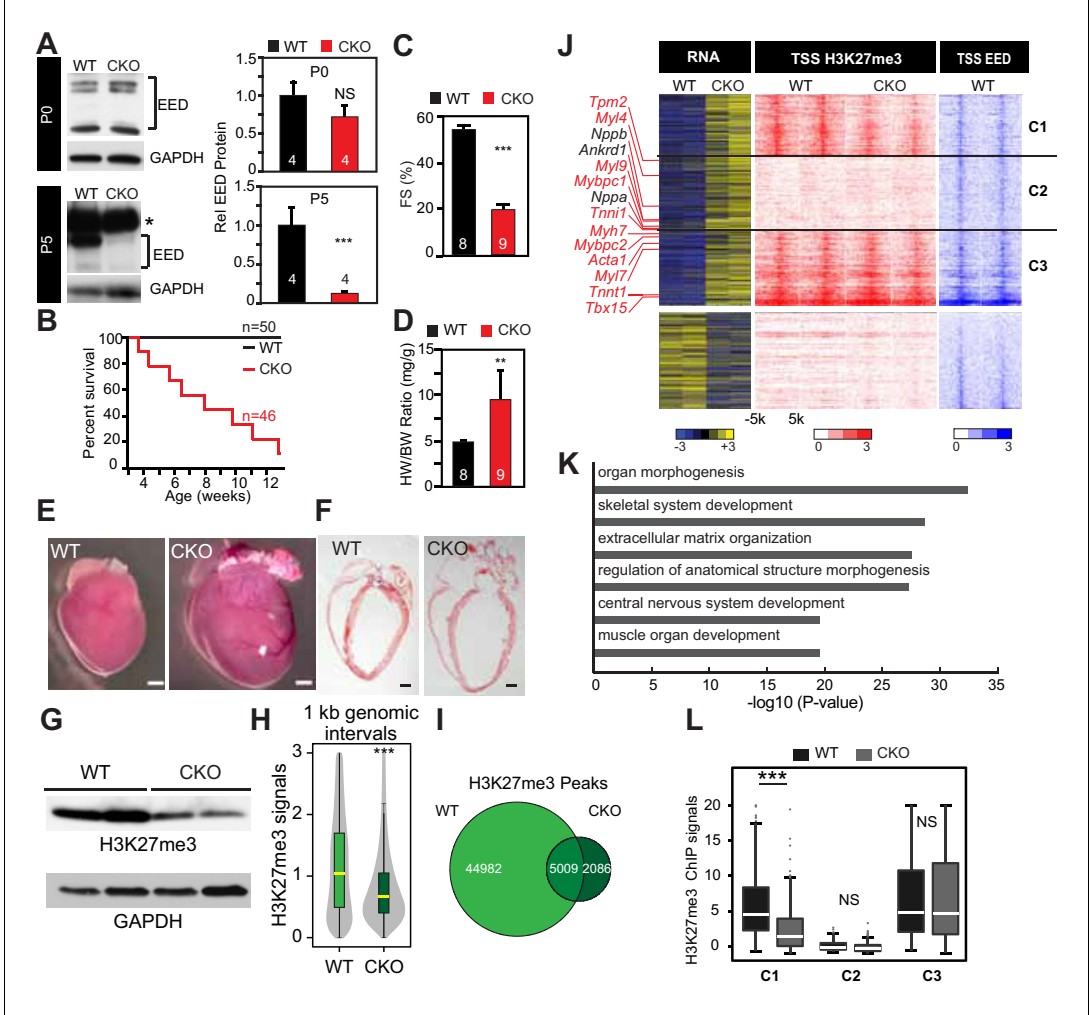

**Figure 1.** Neonatal cardiomyocyte inactivation of *Eed* caused lethal dilated cardiomyopathy. (**A**) EED protein expression in WT and cardiac Eed[CKO] (CKO, Myh6-Cre[+];Eed[f/f]) on postnatal days 0 (P0) and 5 (P5). Quantification shows relative EED protein normalized to GAPDH loading control. Several splice isoforms of EED were detected. * indicates a non-specific band that is larger than full length EED's predicted molecular weight. (**B**) Kaplan-Meier survival curve of WT and Eed[CKO] mice. (**C**) Heart function was measured by echocardiography as fractional shortening (FS%) at 2 months of age. See *Figure 1—figure supplement 2A* for FS% at earlier time points. (**D–F**) Cardiac dilatation and hypertrophy were observed by heart to body weight ratio (**D**), gross morphology (**E**), and histology (**F**) in WT and Eed[CKO] at 2 months of age. Representative hearts are shown. Bar = 1 mm. (**G**) Immunoblotting for H3K27me3 in cardiomyocytes from WT and Eed[CKO] at 2 months of age. (**H**) Genome-wide distribution of H3K27me3 ChIP-seq signals in WT and Eed[CKO] purified cardiomyocytes. ChIP-seq signal was measured in 1 kb windows across the genome. The signal distribution is displayed as a violin plot. Yellow lines denote the median value. (**I**) Venn diagram showing the distribution of H3K27me3 peaks in WT and Eed[CKO] heart. (**J**) Heat map of RNA transcript levels of differentially expressed genes (fold-change >1.5 or <0.67 and adjusted p-value<0.05) are shown in the left heatmap. Expression values for each gene were row scaled. Selected contractile myofiber and heart failure marker genes are shown in red and black, respectively. Right heatmap shows H3K27me3 and EED ChIP-seq signal at the transcriptional start site (TSS) of the differentially expressed gene on the same row. Gene expression, H3K27me3, and EED ChIP-seq studies were performed on purified cardiomyocytes at 2 months of age. Rows were ordered by k-means clustering on H3K27me3 and EED ChIP-seq signal into three clusters, C1-C3. (**K**) Gene Ontology analysis of differentially expressed genes between WT and Eed[CKO]. The top six significant terms are shown. (**L**) Box plots showing H3K27me3 signals in these three clusters as shown in J. A, C, D, Student's t-test; H, L, Wilcoxon-Mann-Whitney test. *p<0.05; **p<0.01; ***p<0.001, NS, not significant. Numbers in bars indicate independent biological replicates.

The following figure supplements are available for figure 1:

**Figure supplement 1.** Eed depletion in WT and EedCKO mice.

**Figure supplement 2.** Characterization of EedCKO mice.

embryonic lethality, but most died over the first 3 months of life (*Figure 1B*). By echocardiography, Eed^CKO mice had left ventricular dilatation and progressive systolic dysfunction (*Figure 1C* and *Figure 1—figure supplement 2A*). Histopathological examination confirmed massive cardiomegaly (*Figure 1D–F*). The expression level of the heart failure marker *Nppa,* encoding atrial natriuretic factor, was strongly upregulated (*Figure 1—figure supplement 2B*). Eed^CKO mice that survived to 2 months of age had substantial CM hypertrophy and cardiac fibrosis (*Figure 1—figure supplement 2C–F*). These results show that *Eed* is essential for neonatal heart maturation and that its inactivation in CMs causes lethal dilated cardiomyopathy.

## Global distribution of H3K27me3 on the promoters/enhancers of derepressed genes

EED is a central, non-redundant subunit of PRC2, an enzyme that canonically represses its target genes by depositing repressive H3K27me3 epigenetic marks (*Margueron et al., 2009*; *Hansen et al., 2008*). As expected, cardiac *Eed* inactivation globally decreased H3K27me3 level in CMs (*Figure 1G*). Genome-wide H3K27me3 occupancy also showed the anticipated global reduction of H3K27me3 signal (*Figure 1H*). Accordingly, *Eed* inactivation reduced the number of regions enriched for H3K27me3 across the genome from 49991 to 7095 (*Figure 1I* and *Supplementary file 1*). Of the 7095 H3K27me3-occupied regions in Eed^CKO CMs, 5009 were also observed in wild-type CMs and 2086 were not (*Figure 1I*). These 2086 new peaks had low intensity, and their centers were distributed as follows: 1040 (49.86%) in intergenic regions, 192 (9.20%) in promoter regions, 721 (34.56%) in introns, 98 (4.70%) in exons, and 35 (1.68%) in 5' and 3' UTR.

To gain insights into the mechanisms underlying the dilated cardiomyopathy phenotype, we performed gene expression profiling of dissociated, purified, 8-week-old CMs using RNA-seq. We identified 863 upregulated and 392 downregulated genes in Eed^CKO (fold change >1.5 or <0.67 and adjusted p-value<0.05; *Figure 1J* and *Supplementary file 1*). Remarkably, we found that slow-twitch myofiber genes, such as *Tnnt1*, *Tnni1*, *Myh7*, *Mybpc1*, *Myl9*, *Myl7*, *Myl4*, *Mybpc2*, and *Acta1*, were significantly upregulated in Eed^CKO heart (*Figure 1J*). Indeed, the upregulated genes were significantly enriched for gene ontology terms related to skeletal muscle genes (*Figure 1K*). Normal cardiac function requires proper expression of cardiac sarcomere genes and repression of skeletal muscle and slow-twitch myofiber genes (*Frey et al., 2000*; *Montgomery et al., 2007*; *Ding et al., 2015*), suggesting that abnormal expression of these genes contributes to weak cardiac contraction in Eed^CKO mice.

To dissect the relationship between the differentially expressed genes and H3K27me3, H3K27ac, and EED occupancy, we performed chromatin immunoprecipitation followed by high-throughput sequencing (ChIP-seq) in purified, 2-month-old CMs. CMs were over 95% pure by immunofluorescence microscopy (*Figure 1—figure supplement 2G*). Markedly reduced H3K27me3 in Eed^CKO CMs demonstrated highly efficient EED inactivation (*Figure 1—figure supplement 2G*). PCR of genomic DNA from purified CMs, as well as review of the RNA-seq track view of the *Eed* locus, further indicated both efficient gene inactivation by Myh6-Cre, which is CM-specific, as well as high CM purity (*Figure 1—figure supplement 2H,I*). ChIP-seq signals were consistent high CM purity. For example, at *Myh*6, expressed in CMs, showed robust signal for the active mark H3K27ac and little signal from the repressive mark H3K27me3 (*Figure 1—figure supplement 2J*). On the other hand, *Vim*, expressed in non-CMs, lacked H3K27ac and was enriched for H3K27me3 (*Figure 1—figure supplement 2K*).

About half of the upregulated genes (49.71%), including 7 of 14 upregulated skeletal muscle genes, were occupied by EED at their TSSs in WT. EED occupancy at a subset of the upregulated genes was confirmed by ChIP-qPCR (*Figure 1—figure supplement 2L*). Unlike upregulated genes, only 8.42% of downregulated genes were bound by EED and H3K27me3. These findings suggest that gene upregulation (derepression) rather than downregulation is the predominant, direct effect of EED deficiency, and that repression of skeletal muscle genes is a key direct role of EED in adult heart.

We evaluated the relationship between H3K27me3 occupancy and gene upregulation in Eed^CKO. First, we quantitatively measured H3K27me3 signal within ±500 bp of TSS on EED target genes in control and Eed^CKO, stratified by H3K27me3 signal in wild type (*Figure 1—figure supplement 2M*). This analysis showed that EED inactivation reduced the median H3K27me3 signal only of the quartile of genes with highest H3K27me3 occupancy in wild type. Surprisingly, the median H3K27me3 signal

in the remaining three quartiles of upregulated genes was not significantly altered by EED inactivation. Consistent with this analysis, the aggregate H3K27me3 signal at the promoters of the 863 upregulated genes was not reduced in Eed[CKO], while it was lower in Eed[CKO] at the unchanged and downregulated genes (*Figure 1—figure supplement 2N*). To further corroborate these findings, we divided upregulated genes into three clusters (*Figure 1J*): those in which upregulation was coupled to H3K27me3 reduction (Cluster C1), those with low H3K27me3 signal (C2), and those with no change in H3K27me3 signal (C3). Cluster C1, which matches the canonical view in which gene derepressed is association with loss of H3K27me3 (*Margueron et al., 2009*; *Hansen et al., 2008*; *Boyer et al., 2006*), contained only 252 of 863 upregulated genes (29%) in Eed[CKO]. In comparison, 4731 of 10,024 expressed genes (47%) had reduced promoter H3K27me3, indicating lack of enrichment of H3K27me3 loss among upregulated genes. Again, the expected relationship between gene upregulation and H3K27me3 loss did not hold for the majority of upregulated genes, as those in Clusters 2 (303 genes) and 3 (308 genes) were not linked to loss of H3K27me3 (*Figure 1L*). Togerther, these analyses indicate that the majority of gene upregulation that occurs with Eed inactivation occurs without loss of H3K27me3. These conclusions were robust to the normalization method used for H3K27me3 ChIP-seq analysis (see Materials and methods), indicating that it was not a result of global differences in H3K27me3 between WT and Eed[CKO]. Furthermore, we independently confirmed retention of H3K27me3 at promoters of selected upregulated genes by ChIP-qPCR (*Figure 1—figure supplement 2O*).

Overall, these results indicate that upregulation of most genes in Eed[CKO], including the derepressed skeletal muscle genes, was not accompanied by loss of H3K27me3, as expected by the canonical model of EED repression via 'writing' of repressive H3K27me3 marks (*Margueron et al., 2009*; *Hansen et al., 2008*).

## Elevated H3K27ac at genes upregulated in Eed[CKO]

Our data suggested that EED repressed gene expression in the postnatal heart through mechanisms other than H3K27me3 deposition. Acetylation of H3K27 (H3K27ac) is linked to gene activation and active enhancers (*He et al., 2014*; *Rada-Iglesias et al., 2011*; *Ferrari et al., 2014*). Increased H3K27ac was previously noted in embryonic stem cells with PRC2 loss of function (*Ferrari et al., 2014*; *Pasini et al., 2010*). Immunoblotting of purified Eed[CKO] CMs showed that *Eed* inactivation markedly increased total histone H4 acetylation and acetylation of histone H3 at K9 and K27 (*Figure 2A*); H3K14ac showed a trend toward being higher in Eed[CKO], although the difference was not statistically significant (p=0.074; *Figure 2A*). This result was unlikely to be an indirect result of cardiac dysfunction in Eed[CKO] hearts, because a similar effect was observed after acute, siRNA-mediated *Eed* knockdown in the HL-1 CM-like cell line (*Figure 2B*).

Genome-wide H3K27ac occupancy analysis by ChIP-seq in 2-month-old WT or Eed[CKO] purified CMs showed that *Eed* inactivation led to acquisition of 8161 new H3K27ac sites and loss of 5950 sites, with 20799 sites in common (*Figure 2C* and *Supplementary file 1*). These peak changes were accompanied by a broad, genome-wide increase of H3K27ac signal strength (*Figure 2D*). Indeed, the aggregate H3K27ac signal at the promoters of the 863 genes upregulated in Eed[CKO] increased, whereas it was either unchanged or reduced for the genes that were unchanged or downregulated, respectively (*Figure 2E*). These results were validated by ChIP-qPCR at the promoters of several slow twitch muscle genes that were upregulated in Eed[CKO] (*Figure 2—figure supplement 1*). Increased H3K27ac signal was also observed at enhancer regions (H3K27ac peaks region in either WT or Eed[CKO]) associated with these upregulated genes (*Figure 2E*). Next, we further examined the specific TSS regions for the correlation of H3K27ac change with gene upregulation (*Figure 2F*; rows ordered as in *Figure 1J*). Heatmap analysis of all three clusters and quantitative analysis in box plots for each cluster confirmed that all 863 derepressed genes markedly gained H3K27ac levels (*Figure 2F,G*). Together these data suggest that EED is required to suppress H3K27ac, and that loss of this function leads to abnormal H3K27ac accumulation, global increase in histone H3 and H4 acetylation, and aberrant gene upregulation.

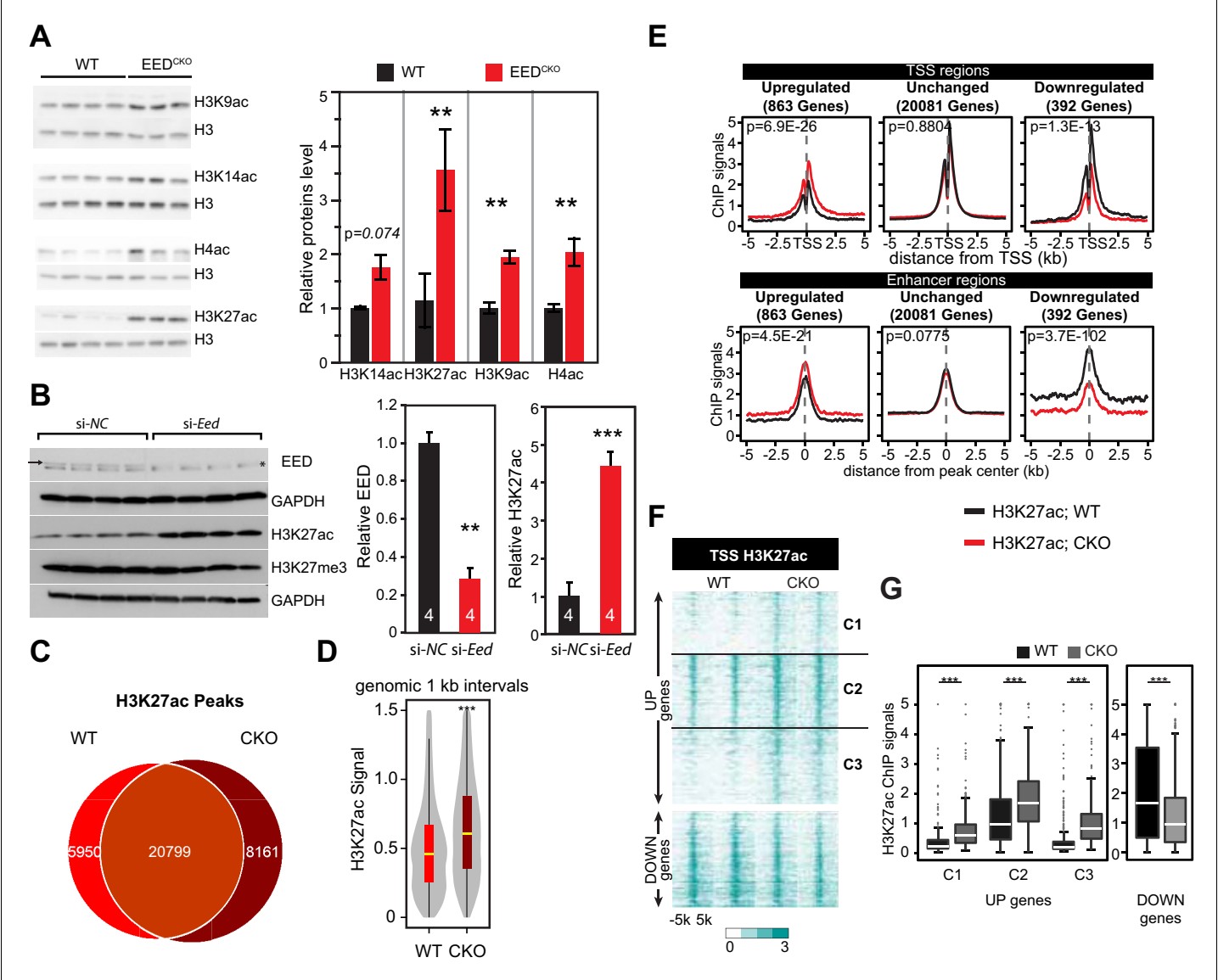

**Figure 2.** *Eed* depletion induced globally elevated histone acetylation. (**A**) Global upregulation of histone H3 and H4 acetylation at different lysine residues in isolated adult cardiomyocytes from 2-month-old WT and Eed[CKO] hearts. Histone levels were measured by immunoblotting and further quantified by normalization to total histone H3. (**B**) Acute EED depletion increased H3K27ac levels in HL-1 cardiomyocyte-like cells. Fully confluent HL-1 cells were transfected with TriFECTa DsiRNAs against Eed (si-Eed) or scrambled sequence-negative control (si-NC). Protein levels were measured by quantitative immunoblotting. Arrow, EED band. Asterisk, non-specific band. (**C**) Venn diagram showing the overlap of H3K27ac ChIP-seq peaks in isolated cardiomyocytes from WT and Eed[CKO] hearts at 2 months of age. (**D**) Genome-wide distribution of H3K27ac signals. The violin plot displays H3K27ac ChIP-seq signals in 1 kb windows across the genome. Yellow horizontal lines denote median values. (**E**) Aggregation plots of H3K27ac ChIP-seq signals at ±5 kb of TSS (upper row) or at distal regions (lower row) of genes that were upregulated, downregulated, or unchanged by *Eed* inactivation. (**F–G**) Heat map (**F**) and box plots (**G**) of H3K27ac levels at TSS of differentially expressed genes. The row order and clustering is the same as in *Figure 1J*. A, B, Unpaired Student's t-test; D,G, Wilcoxon-Mann-Whitney test. **p<0.01; ***p<0.001; NS, not significant.

The following figure supplement is available for figure 2:

**Figure supplement 1.** H3K27ac ChIP-qPCR validation.

## Reintroduction of EED normalizes heart function and reinstates H3K27ac but not H3K27me3

Additional data from timed EED replacement studies further strengthened the relationship between the Eed[CKO] phenotype and H3K27ac rather than H3K27me3. EED has been proposed to facilitate perpetuation of H3K27me3 marks by itself binding to H3K27me3, thereby reinforcing deposition of H3K27me3 at existing sites (*Margueron et al., 2009*; *Hansen et al., 2008*). If this is the case, then ablation of EED should disrupt the H3K27me3 landscape, preventing its restoration even if EED is later re-expressed. To test this model, we temporally re-expressed EED in CMs by taking advantage of highly efficient, durable, and CM-selective gene transfer using adeno-associated virus serotype 9 (AAV9) and the cardiac troponin T promoter (*Tnnt2*) (*Lin et al., 2015*; *Jiang et al., 2013*). We developed AAV9-Tnnt2-EED (abbreviated AAV9-EED) and validated that it drove cardiac expression of EED when delivered to mice at P14 or P25 (*Figure 3—figure supplement 1A–C*). We evaluated the effect of delayed EED re-expression at either P14 or P25 (*Figure 3A*). Interestingly, delayed EED re-expression was still able to rescue cardiac systolic function and cardiac hypertrophy, normalize expression of myofiber and heart failure genes, and ameliorate cardiomegaly of Eed[CKO] mutants (*Figure 3B–D* and *Figure 3—figure supplement 1D*).

Next, we investigated the effect of delayed EED re-expression on the chromatin landscape. Delayed EED re-expression corrected the abnormally high global H3K27ac levels observed in Eed[CKO] (*Figure 3E*). However, delayed EED re-expression did not correct the low H3K27me3 levels seen in Eed[CKO] (*Figure 3E*). This result was reinforced by genome-wide measurement of H3K27me3 and H3K27ac chromatin occupancy by ChIP-seq. Whereas EED re-expression at P14 and P25 both conferred phenotypic rescue and normalized the genome-wide distribution of H3K27ac signal, P14 and P25 EED re-expression did not normalize H3K27me3 distribution (*Figure 3F,G*).

We then evaluated the effect of EED re-expression on the 863 genes derepressed in CKO-luc by performing RNA-seq on purified CMs. Out of 863 genes that were upregulated in Eed[CKO], 256 (66, 87, and 103 in Cluster 1, 2, and 3, respectively) were significantly downregulated (*Figure 3F*). This normalization of gene expression was associated with reduction of H3K27ac, but not H3K27me3, near the TSS (*Figure 3F,G*). Inspection of the *Acta1* and *Myl7* loci reinforced the finding that, at the upregulated slow twitch skeletal muscle genes, delayed EED re-expression corrected abnormal deposition of H3K27ac but did not significantly affect H3K27me3 (*Figure 3H*). These data are consistent with the model that EED 'reads' its own mark to reinforce its 'writing', so that proper writing cannot be restored after a period has elapsed during which the mark decays. An alternative explanation is that CM PRC2 histone trimethylase activity declines with age due to downregulation of EZH2 and lack of detectable trimethylase activity from EZH1-containing complexes from adult CMs. Regardless of the precise mechanism, our data show that phenotypic rescue and gene expression correction by AAV-EED were not associated with normalization of H3K27me3. Rather, correction aligned with normalization of H3K27ac epigenetic marks.

## EED interacts with HDACs and enhances its deacetylase activity but is dispensable for HDAC recruitment

Histone H3 and H4 acetylation, including H3K27ac, is regulated by a balance between histone acetyltransferases and histone deacetylases (HDACs) (*Stefanovic et al., 2014*; *Tie et al., 2009*; *Wang et al., 2009*). HDACs generally function as co-repressors, and class I and class II HDACs have critical roles in regulating cardiac gene expression and function, including suppression of skeletal muscle gene expression (*Montgomery et al., 2007*; *Song et al., 2006*; *Zhang et al., 2002*; *Trivedi et al., 2007*). Although Eed[CKO] hearts had elevated global histone H3 and H4 acetylation, most strikingly inducing H3K27ac, total HDAC levels were not changed (*Figure 4—figure supplement 1A,B*). Otte and colleagues showed that HDAC2 interacts with EED and that EED co-immunoprecipitated protein complexes containing HDAC activity (*van der Vlag and Otte, 1999*), suggesting that HDACs participate in EED-mediated gene repression. These lines of evidence led us to hypothesize that EED represses its postnatal cardiac target genes by interacting with HDACs to decrease their H3K27ac. To test this hypothesis, we screened class I and class II HDACs (HDAC1-HDAC9) for interaction with EED. Immunoprecipitation of FLAG-tagged HDAC1-HDAC9 co-precipitated HA-tagged EED, with the exception of HDAC8 (*Figure 4A* and *Figure 4—figure supplement 1C*). This was confirmed by reciprocal immunoprecipitation experiments, in which HA-EED co-

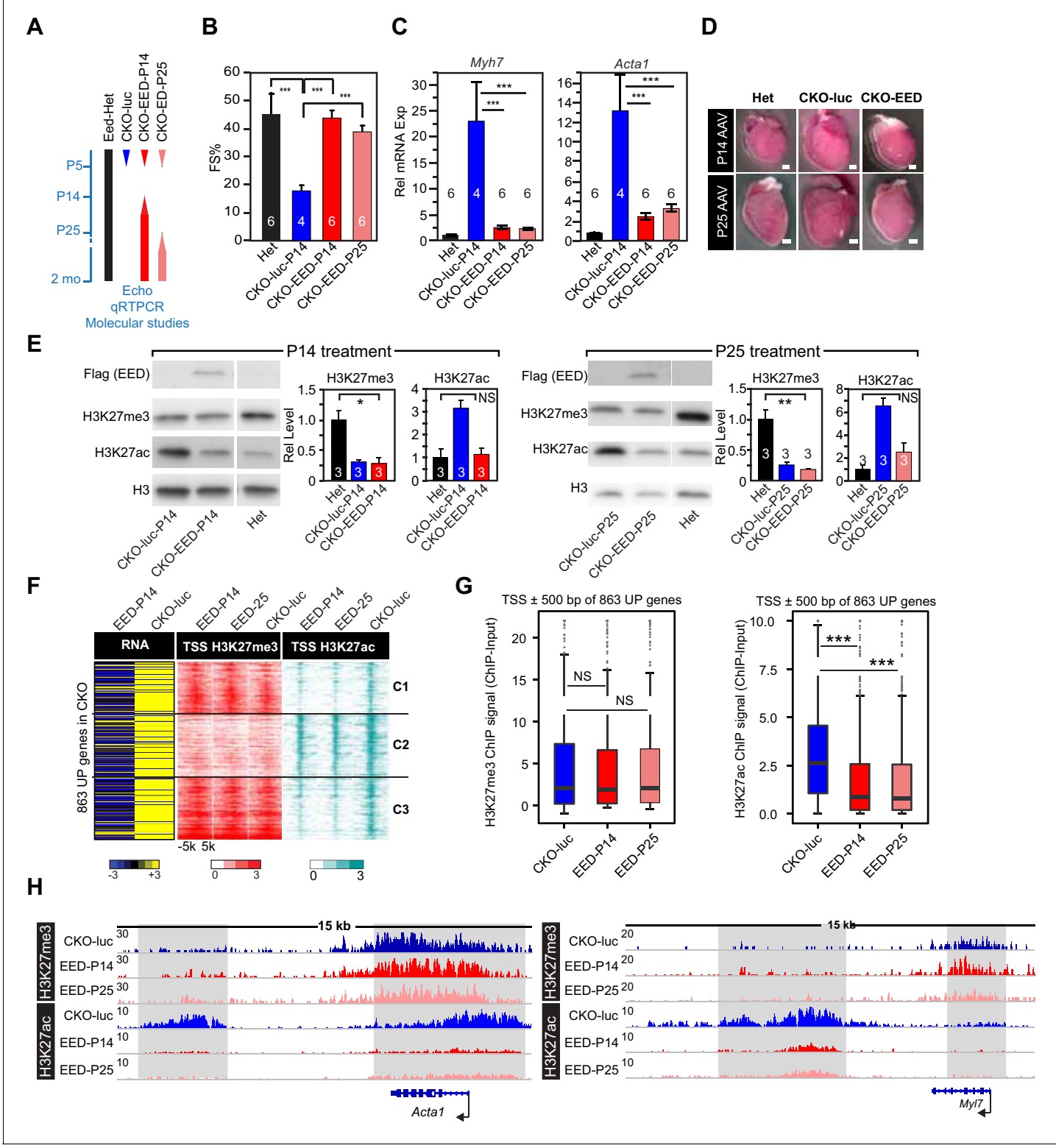

**Figure 3.** Delayed re-expression of EED rescued heart function and normalized H3K27ac but not H3K27me3. (**A**) Schematic of the experimental design. AAV9 expressing EED or luciferase (luc) in cardiomyocytes was injected at P14, or P25 to control or Eed$^{CKO}$ mice. Lines represent the temporal pattern of EED expression. (**B–E**) Heterozygous (Het; Eed$^{fl/+}$; Myh6Cre$^+$) and Eed$^{CKO}$ mice were injected with AAV-luc or AAV-EED at indicated ages. At 2 months of age, heart function (FS%) was measured by echocardiography (**B**), and expression of *Myh7* and *Acta1*, two slow-twitch myofiber genes aberrantly expressed in Eed$^{CKO}$, were measured by qRTPCR (**C**). Representative images showing gross morphology of hearts at 2 months of age (**D**).
*Figure 3 continued on next page*

*Figure 3 continued*

Cardiomegaly of CKO-luc hearts was abrogated by EED replacement at P14 or P25. Isolated cardiomyocytes were immunoblotted to measure expression of virally delivered EED, and global levels of H3K27ac and H3K27me3 (E) Graphs show quantitation of global H3K27ac and H3K27me3 levels, normalized to histone H3. (F) Heatmaps of RNA expression, H3K27me3 and H3K27ac ChIP signals at ±5 kb of TSS of 863 upregulated genes from Eed$^{CKO}$ mice injected with AAV-luc or AAV-EED at P14 or P25. Row order and cluster labels are the same as *Figure 1J*. (G) Quantitative analysis of H3K27me3 and H3K27ac ChIP signals near TSSs shown in F by box plots. (H) Genome browser view of H3K27me3 and H3K27ac ChIP-seq signals at the *Acta1* or *Myl7* loci. The regions highlighted in gray show that gain of H3K27ac in Eed$^{CKO}$ was reset to normal under EED rescue conditions regardless of H3K27me3 status. B, C, and E, Unpaired Student's t-test; G, Wilcoxon-Mann-Whitney test. *p<0.05; **p<0.01; ***p<0.001.

The following figure supplement is available for figure 3:

**Figure supplement 1.** AAV9-EED rescue of EEDCKO mice.

---

precipitated HDAC1-HDAC9 except HDAC8 (*Figure 4A* and *Figure 4—figure supplement 1D*). We further confirmed the interaction of endogenous EED and HDAC2 in HL-1 CM-like cells (*Figure 4B*).

To further approach the molecular function of this interaction on a genome-wide level, we examined the genomic binding distribution of EED and HDAC2 in adult CMs. Nearly half of HDAC2 peaks (4042 out of 9869, 41%) overlapped with EED peaks in WT, in line with their biochemical interaction (*Figure 4C*). HDAC2, H3K27me3, and H3K27ac signal at the entire set of 8625 EED occupied regions further confirmed this result (*Figure 4D*). We identified three classes of EED-bound genomic domains (*Figure 4D*). A first class (E1, 5073 regions) was marked by EED, HDAC2 and H3K27ac, but had little H3K27me3. This class was consistent with a previous study that showing that HDACs mark not only inactive genes but also active genes that are poised for transcriptional repression (*Wang et al., 2009*). A second class (E2, 1830 regions) had strong EED and H3K27ac signals, and H3K27me3 was present but weak. A third class (E3, 1722 regions) was marked by strong and broad EED and H3K27me3, but little H3K27ac. Together, these data suggest that EED may function in a complex involving HDACs, independent of H3K27me3.

Overall, HDAC2 signals did not change in EED deficient CMs (*Figure 4E*). This finding was also confirmed by ChIP-qPCR measurement of HDAC2 or HDAC5 enrichment at promoters of genes upregulated in Eed$^{CKO}$ compared to control, revealing no significant difference between HDAC2 or HDAC5 occupancy (*Figure 4F* and *Figure 4—figure supplement 1E*). Given that upregulated genes accounted for a small portion of EED occupied regions, we further analyzed HDAC2 occupancy of genes upregulated, unchanged or downregulated in Eed$^{CKO}$. This analysis reinforced that Eed inactivation did not change HDAC2 occupancy, indicating that HDAC2 accumulation on genomic domains is independent of EED (*Figure 4G*). Thus, elevated acetylation on H3K27ac in Eed$^{CKO}$ was unlikely due to decreased HDAC binding.

To identify mechanisms that account for increased H3K27ac at upregulated, EED-occupied genes in Eed$^{CKO}$ CMs, we considered the possibility that EED biochemically regulates HDAC activity. We measured the in vitro deacetylase activity of HDAC2 in the presence of increasing amounts of recombinant EED protein, expressed and affinity purified from S*f*21 cells using a baculovirus expression system. EED protein did not contain detectable contaminating proteins (*Figure 4—figure supplement 2A*). EED alone had no detectable deacetylase activity, but when added to HDAC2 it augmented HDAC2's deacetylase activity by up to threefold (*Figure 4H*). Together, the aforementioned data all support a model in which EED represses a subset of its target genes by stimulating HDAC. These results also raise the possibility that EED drives transcriptional repression at least partially by modulating the activity of another class of epigenetic repressors, HDACs.

## HDACs functions downstream of EED to direct normal heart maturation

Both class I and class II HDACs regulate cardiac gene expression, hypertrophy, and function (*Hohl et al., 2013*; *Hang et al., 2010*; *Trivedi et al., 2007*; *Kong et al., 2006*; *Kook et al., 2003*; *Zhang et al., 2002*). Notably, cardiac-restricted deletion of both HDAC1 and HDAC2 caused dilated cardiomyopathy accompanied by upregulation of genes encoding skeletal muscle-specific contractile proteins (*Montgomery et al., 2007*). These findings converged with our studies to suggest the hypothesis that EED inactivation caused cardiomyopathy and upregulation of skeletal muscle myofiber genes through loss of HDAC1/2 activity. To test this hypothesis, we asked if overexpression of

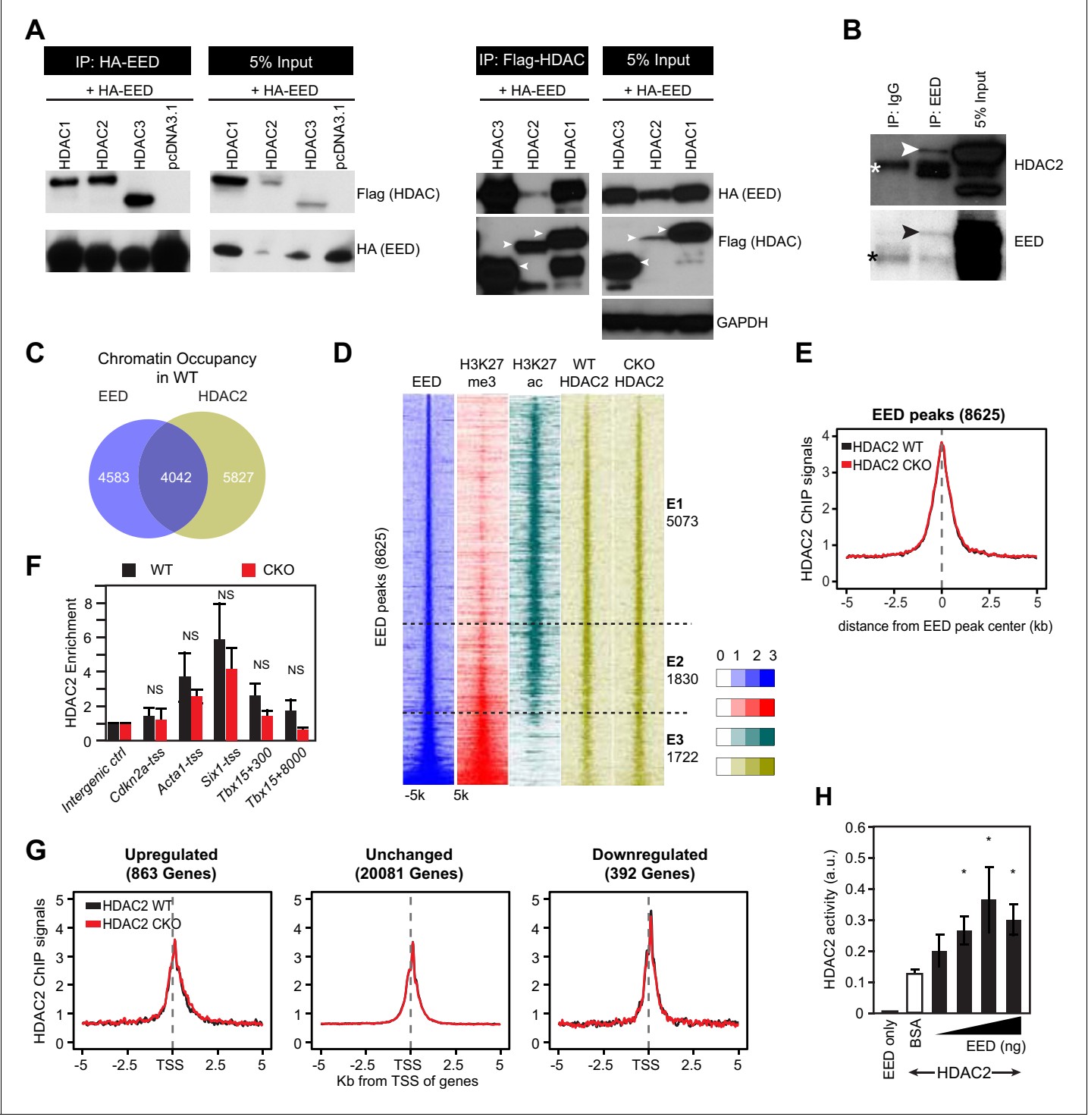

**Figure 4.** EED interacts with and co-localizes with HDAC to repress transcription through enhancing its deacetylation activity. (**A**) Co-Immnoprecipitation analysis of EED-HDAC interaction in 293 T cells. HA-EED pull down with HA antibody co-precipitated FLAG-HDAC1/2/3 (left). Reciprocally, FLAG-HDAC1/2/3 pull down with Flag antibody co-precipitated HA-EED (right). Data on HA-EED and Flag-HDAC4-9 interaction are presented in **Figure 4—figure supplement 1C,D**. (**B**) Interaction between endogenous EED and HDAC2. EED, immunoprecipitated from HL-1 cardiomyocyte-like cells, co-precipitated HDAC2. Arrowhead denotes the specific band and asterisk denotes the non-specific IgG heavy chain band. (**C**) Venn diagram showing overlap of EED and HDAC2 peaks in WT. (**D**) Heatmaps showing ChIP-seq signals for EED, H3K27me3, H3K27ac and HDAC2 at ±5 kb of EED peak centers. Rows were sorted by ascending EED peak signal. (**E**) Aggregate plot showing HDAC2 ChIP signals in WT and Eed[CKO], centered on EED peak centers. (**F**) HDAC2 occupancy of the indicated chromatin regions in isolated cardiomyocytes from WT and Eed[CKO] mice at 2

*Figure 4 continued on next page*

*Figure 4 continued*

months of age. Occupancy was measured by ChIP followed by quantative PCR (ChIP-qPCR). Chromatin regions are named by the adjacent gene and the distance to the TSS. (G) Aggregation plot showing HDAC2 ChIP-seq signals in WT and Eed^CKO at ±5 kb of TSS of genes that were upregulated, unchanged or downregulated in Eed^CKO. (H) Effect of EED on HDAC2 activity. In vitro deacetylation assay was performed using recombinant active HDAC2 (50 ng) in the presence of BSA or 5 to 100 ng of recombinant EED, purified from insect cells. Deacetylation activity was measured by colorimetric assay. F, H, I, J, K, Unpaired Student's t-test. *p<0.05; **p<0.01; ***p<0.001.

The following figure supplements are available for figure 4:

**Figure supplement 1.** HDAC-EED interaction.

**Figure supplement 2.** Validation of HDAC2 and EED proteins purity and dCas9-EED interaction with EZH2.

HDAC1/2 ameliorated the Eed^CKO phenotype. Accordingly, we developed AAV9-Tnnt2-HDAC1/2 (abbreviated AAV-HDAC1/2) to direct CM-selective overexpression of HDAC1/2 (*Figure 5—figure supplement 1A–C*). Remarkably, AAV-HDAC1/2 delivery to P3 Eed^CKO mice (CKO-HDAC1/2) normalized systolic dysfunction and cardiomegaly, compared to control treatment with luciferase (CKO-luc; *Figure 5A–C*). Heterozygous littermates (Het; *Eed*^fl/+; *Myh6*^Cre) were used as unaffected controls. AAV-HDAC1/2 also restored repression of slow twitch sarcomere genes *Acta1* and *Myh7*, and reduced expression of the heart failure marker *Nppa* (*Figure 5D*). RNA-seq gene expression measurements showed that out of 863 genes upregulated in Eed^CKO, 104 were downregulated in the rescue group (adjusted p-value<0.05 and fold change (CKO-luc/CKO-HDAC1/2) >1.5; *Figure 5E*). Of these 104 genes, 39, 34, and 31 were found in Clusters 1, 2, and 3, respectively.

We measured the effect of AAV-HDAC1/2 on genome-wide chromatin occupancy of H3K27me3 and H3K27ac by ChIP-seq (*Figure 5E*). This revealed that the greater H3K27ac signal observed at the TSS of most of the 863 genes upregulated in Eed^CKO was normalized by AAV-HDAC1/2: the ratio of H3K27ac signal in CKO-luc to CKO-HDAC1/2 was ≥1.5 for 460 genes and 1–1.5 for 293 genes, while only 110 were not changed (*Figure 5E*). This conclusion was reinforced by comparison of H3K27ac signal between CKO-HDAC1/2 and CKO-luc (*Figure 5F*). Within each of the three classes of regions (C1, C2, C3) defined previously based on H3K27ac occupancy and change with Eed inactivation (*Figure 1J*), AAV-HDAC1/2 significantly reduced H3K27ac signal. Similar results were observed when all H3K27ac regions that increased in EED loss of function were analyzed, rather than only the TSSs of upregulated genes (*Figure 5—figure supplement 1D–F*). Normalization of H3K27ac was further supported by calculating each upregulated gene's change in H3K27ac at the TSS in *Eed* loss of function (CKO-luc/het) compared to its change with AAV-HDAC1/2 rescue (CKO-luc/CKO-HDAC1/2; *Figure 5—figure supplement 1F*, upper panel). For the 104 normalized genes, H3K27ac occupancy was reduced toward control levels. In contrast, no similar change was observed for H3K27me3 (*Figure 5—figure supplement 1F*, lower panel).

Notably, these changes in H3K27ac occurred despite lack of global H3K27ac level, as measured by immunoblotting (*Figure 5—figure supplement 1G*). Thus, heart function normalization was not dependent upon broad normalization of H3K27ac but rather on targeted HDAC1/2 activity at specific loci. On the other hand, H3K27me3 was not strongly affected by AAV-HDAC1/2 (*Figure 5E–G* and *Figure 5—figure supplement 1D–G*), consistent with the findings in EED rescue experiments that correction of gene expression and heart function was independent of H3K27me3.

## HDAC inhibition antagonized Eed^CKO rescue by EED re-expression

To further probe the requirement of HDAC for EED activity in CMs, we studied the effect of HDAC inhibition on rescue of Eed^CKO by AAV-mediated EED re-expression. Neonatal Eed^CKO mice were treated with AAV9-EED at P5. They were then treated daily with the broad-spectrum HDAC inhibitor suberanilohydroxamic acid (SAHA) or vehicle control. Het mice transduced with AAV9-luc and then SAHA or vehicle served as additional controls. Consistent with previous reports (*Xie et al., 2014b*), SAHA treatment for 2 months did not adversely effect heart function in the Het controls, nor did it perturb expression of 5 genes (*Acta1, Myl9, Tnni1, Tbx15* and *Mybpc2*) that were aberrantly upregulated in Eed^CKO. Interestingly, SAHA treatment appeared to block heart function rescue by AAV9-EED re-expression. Two out of five mice died prematurely in the Eed^CKO+EED+SAHA group and

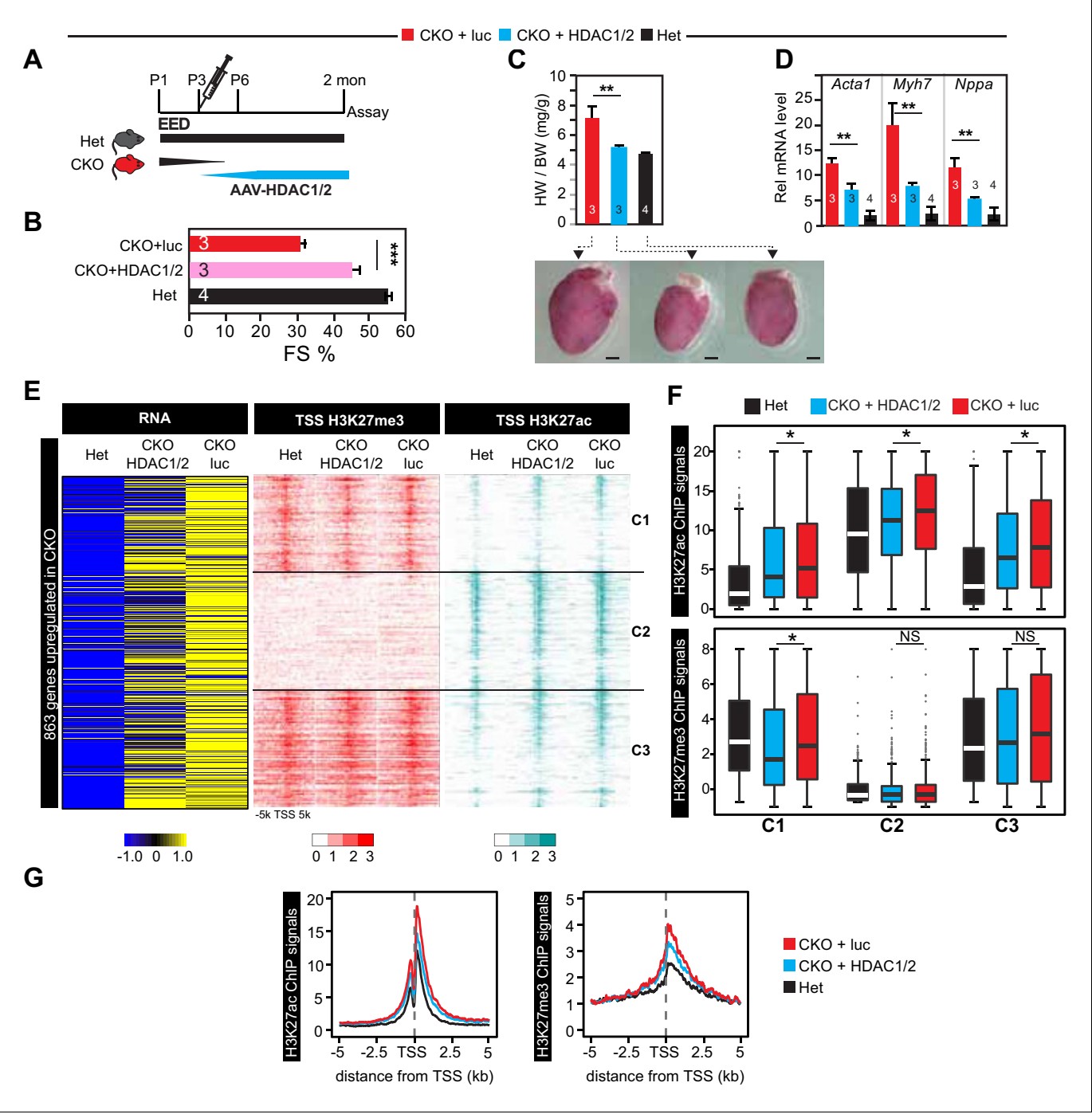

**Figure 5.** Re-introduction of HDAC1/2 restored normal heart function. (**A**) AAV-HDAC1/2 rescue of Eed[CKO]. Schematic shows the rescue experiment design. WT and CKO pups were injected with AAV-luc or AAV-HDAC1/2 at P3, and assays were done at 2 months of age. (**B**) Heart function (FS%) was measured at 2 months of age by echocardiography. (**C**) Gross morphology and heart to body weight ratio of hearts from mice at 2 months of age. Bar = 1 mm. (**D**) Analysis of heart failure gene expression. *Acta1*, *Myh7*, and *Nppa* levels in isolated cardiomyocytes were measured by qRTPCR. (**E, F**) Heatmaps (**E**) and box plots (**F**) showing ChIP signals for H3K27ac and H3K27me3 at ±5 kb of TSS of genes upregulated in Eed[CKO]. Row order and cluster labels are the same as *Figure 1J*. Comparative analysis of ChIP-seq signals was performed within each cluster as indicated. (**G**) Aggregation plots for H3K27me3 or H3K27ac ChIP-seq signals at ±5 kb of TSS of Eed[CKO] upregulated genes in Het, CKO+luc, and CKO+HDAC1/2 groups. B,C,D, Unpaired Student's t-test. F, Wilcoxon-Mann-Whitney test. *p<0.05; **p<0.01; ***p<0.001.

The following figure supplement is available for figure 5:

*Figure 5 continued on next page*

*Figure 5 continued*

**Figure supplement 1.** Effect of over-expression of HDAC1/2 on genome-wide H3K27ac and H3K27me3 accumulation at H3K27ac peaks with increased signal in EedCKO.

were lost to the study. Despite this potential survivor bias, heart function rescue by AAV9-EED tended to impair AAV9-EED rescue in the remaining three mice, although this did not reach statistical significance (p=0.07; *Figure 6A,B*). These data suggest that HDAC activity is essential to suppress aberrant gene expression in Eed$^{CKO}$. In line with this, normalized transcription of four out of five genes by EED re-expression in Eed$^{CKO}$ was blocked by SAHA treatment (*Figure 6C*). The different effect of SAHA on normal hearts compared to AAV9-EED rescue of EED deficiency suggests that multiple redundant repressive pathways operate in normal heart development, making the HDAC pathway dispensable. However, in AAV-EED rescue of EED deficiency, these redundancies are disrupted, perhaps as a result of perturbations of the chromatin landscape, leaving heart development vulnerable to HDAC inhibition. Collectively, these data indicate that normalization of HDAC activity by EED re-expression was essential for rescue of heart failure in Eed$^{CKO}$.

Altogether, these data demonstrate that HDACs collaborate with EED to mediate postnatal cardiac gene repression required for normal heart development and function. Furthermore, our data

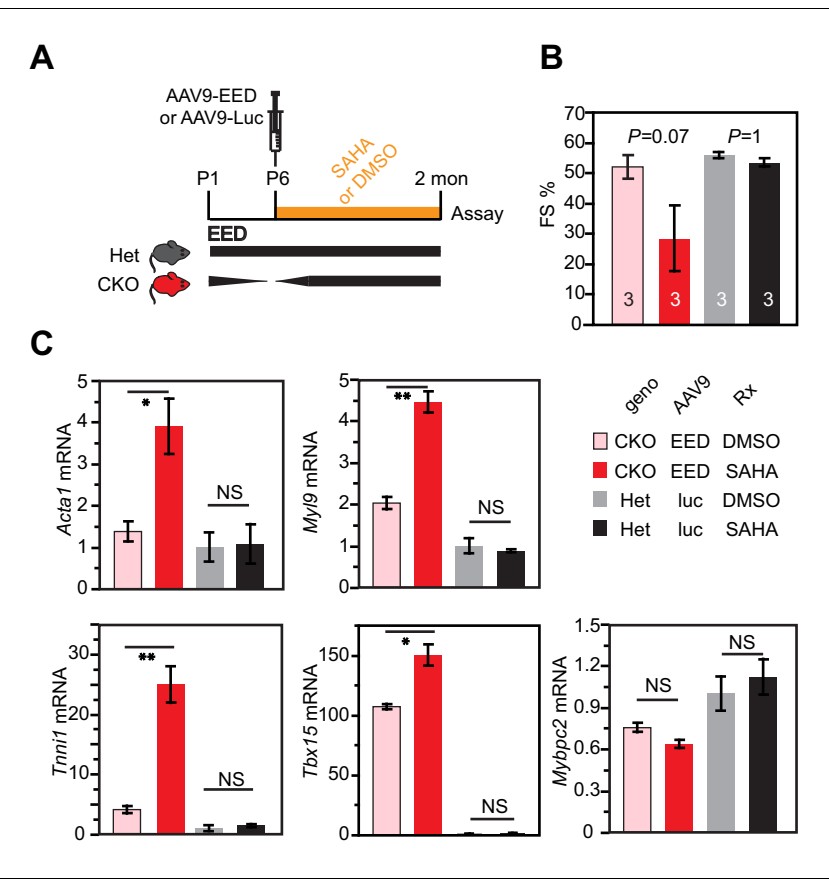

**Figure 6.** HDAC inhibition antagonized rescue of Eed$^{CKO}$ by EED re-expression. (**A**) Schematic of the experimental design. (**B**) Heart function was measured by echocardiography as fraction shortening (FS%) in 2-month-old Eed$^{CKO}$ or Het mice that received the indicated treatments. Two of five Eed$^{CKO}$ + EED + SAHA mice died prior to the study endpoint and were not available for echocardiography. (**C**) qRTPCR measurement of five selected genes that were aberrantly expressed in Eed$^{CKO}$ hearts. P-value by Student's t-test. *p<0.05; **p<0.01; NS, not significant.

indicate that EED functions genetically upstream of HDACs and support a model in which EED maintains transcriptional repression of target genes in post-mitotic cardiomyocytes. Our results demonstrate an essential mechanism of EED repression that is independent of its requirement for PRC2 deposition of H3K27me3 (*Figure 7*).

## Discussion

We found that postnatal loss of EED caused lethal dilated cardiomyopathy associated with ecoptic expression of a slow skeletal muscle gene program. Surprisingly, de-repression of this program was not associated with loss of H3K27me3 at these genes, but rather marked gain of H3K27ac. Moreover, EED depletion caused global increase in histone H3 and H4 acetylation. Our rescue experiments further demonstrated that phenotypic reversal was associated with normalization of H3K27ac but not H3K27me3. Furthermore, we demonstrate a physical and genetic interaction between EED and HDACs that mechanistically accounts, at least in part, for the link between EED loss of function and aberrant H3K27ac accumulation. Together these experiments highlight an H3K27me3-independent repressive activity of EED, and demonstrate that disease caused by aberrant epigenetic regulation can be reversed through epigenetic rewiring.

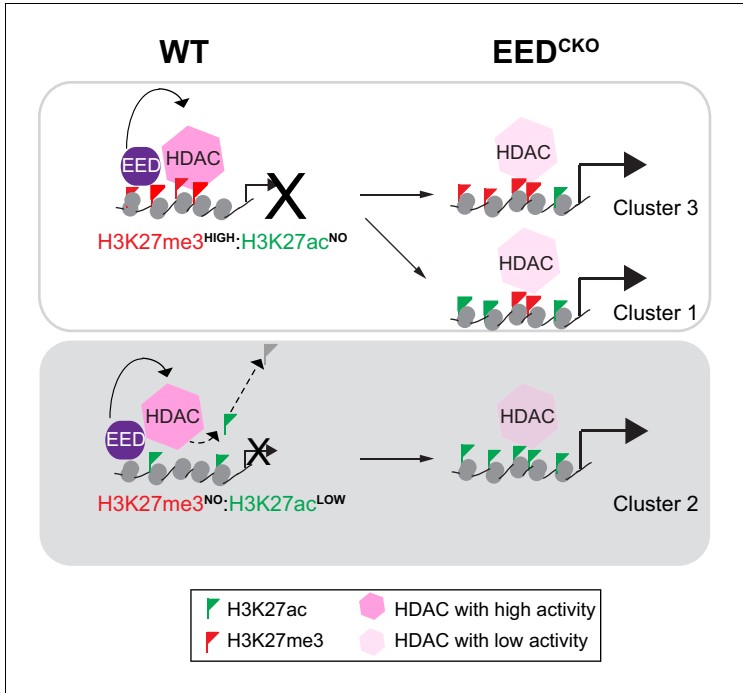

**Figure 7.** Working model delineates a non-canonical mechanism by which EED represses gene transcription. Two mechanisms for EED repression were operative in the postnatal heart. One subset of repressed genes was occupied by EED and H3K27me3 in WT, and EED inactivation reduced H3K27me3 in association with gain of H3K27ac. Upregulation of these genes in Eed^CKO could be due to a combination of loss of H3K27me3 (canonical mechanism). Loss of EED itself with subsequent gain in H3K27ac might also contribute to regulation of these genes. A second subset of repressed genes was also occupied by EED and H3K27me3 in WT, but H3K27me3 was not reduced by EED inactivation. While H3K27me3 may contribute to the repression of these genes in WT, their upregulation in Eed^CKO was not attributable to H3K27me3, which was unchanged. Rather, our data suggest that upregulation was directly due to loss of EED itself, with consequent reduction of HDAC activity and gain in H3K27ac. A third subset of genes was occupied by EED but little H3K27me3 in WT. These genes had significant H3K27ac at baseline, which was further increased by Eed inactivation. Thus, these genes may represent a set 'poised' for activation; in WT EED occupancy represses these genes by collaborating with HDAC to limit gene activity. EED inactivation reduces HDAC activity, resulting in H3K27ac accumulation and gene upregulation.

## Gene de-repression in Eed[CKO] was linked to gain of H3K27ac and not loss of H3K27me3

Previous studies of PRC2 inactivation, performed primarily in cultured cells such as embryonic stem cells, have shown that many de-repressed genes are associated with loss of H3K27me3 (*Simon and Kingston, 2013*, *2009*). The role of PRC2 methylation of H3K27me3 in gene repression was further reinforced by studies of Drosophila imaginal discs in which wild-type histone H3 was replaced by a point mutation, H3K27R, that can neither be methylated nor acetylated (*Pengelly et al., 2013*). The effect of this H3K27R mutation resembled PRC2 inactivation, suggesting that in this system the major repressive activity of PRC2 is trimethylation of H3K27. However, several lines of evidence from our study indicate that an essential mechanism of PRC2 repression in adult CMs is independent of H3K27me3. First, genes de-repressed by *Eed* inactivation had largely unchanged H3K27me3. This result was robust to the method used to normalize the H3K27me3 ChIP-seq data and was independently validated at selected sites by ChIP-qPCR. Second, AAV-EED rescued heart function without correcting global H3K27me3 levels or occupancy of upregulated gene TSSs. Third, AAV-HDAC1/2 rescued heart function and re-established gene repression without elevating H3K27me3.

Our data indicate that in adult CMs, PRC2 regulation of H3K27ac plays an essential role in repression of a subset of genes. First, gene upregulation in Eed[CKO] was associated with gain of H3K27ac. Second, AAV-EED rescue correlated with restoration of H3K27ac rather than H3K27me3. Third, AAV-HDAC1/2 rescued function of Eed[CKO] hearts and normalized H3K27ac but not H3K27me3. HDACs have multiple substrates, including other histone acetylation sites as well as non-histone proteins. Our data do not exclude important effects of HDACs on targets other than H3K27ac, but the close correlation between H3K27ac and cardiac rescue strongly suggests that HDAC acts, at least in part, through H3K27ac. Depletion of essential PRC2 subunits in ES cells and *Drosophila* was previously reported to increase H3K27ac (*Pasini et al., 2010*; *Tie et al., 2009*; *Ferrari et al., 2014*), consistent with the genetic interplay between PRC2 and histone acetylation that we observed in this study. Similarly, dominant negative inhibition of PRC2 by expression of a point mutant of histone H3.3 (H3.3K27M) globally increased H3K27ac (*Herz et al., 2014*). Furthermore, EED-HDAC2 interaction and HDAC activity were previously reported to be essential for PRC2 repression of selected target loci (*van der Vlag and Otte, 1999*). However, the global role of H3K27me3 versus H3K27ac in EED-mediated repression in vivo has remained uncertain. Our study clarifies this relationship in postnatal CMs, by demonstrating that gene de-repression downstream of EED loss of function is closely tied to H3K27ac gain rather than to H3K27me3 loss.

As reviewed above, the canonical dogma is that PRC2 represses genes by depositing H3K27me3. Why might our study have reached a different conclusion? One key difference is the mitotic state of the model systems: whereas the vast majority of studies of PRC2 have been performed in actively cycling cells such as embryonic stem cells and developing embryos, mature CMs have largely exited the cell cycle. Cycling cells require active rewriting of histone marks, so that PRC2 inactivation is rapidly accompanied by widespread H3K27me3 loss. In contrast, adult CMs have relatively stable H3K27me3 and little detectable PRC2 methyltransferase activity. Under these conditions, there is likely a greater opportunity to expose writing-independent activities of EED. Importantly, many cells of adult mammals are either slowly cycling or terminally differentiated, and thus aspects of their epigenetic regulation may more closely resemble adult CMs than cultured cells such as embryonic stem cells.

## Remodeling the chromatin landscape to correct heart failure

Mutations of epigenetic regulators cause congenital heart disease, and abnormalities of the chromatin landscape have been implicated in the pathogenesis of heart failure (*He et al., 2012b*; *Zaidi et al., 2013*; *Hang et al., 2010*; *Delgado-Olguín et al., 2012*; *Lickert et al., 2004*; *Montgomery et al., 2007*). The chromatin landscape is not hardwired but rather dynamically constructed through the sequential action of epigenetic regulators, suggesting that perturbation of the normal sequence may irreversibly disrupt the chromatin landscape. Consistent with this expectation, we found that transient loss of EED irrevocably altered the landscape of H3K27me3. EED inactivation also globally increased H3K27ac levels, and gene de-repression was associated with H3K27ac gain on TSS regions of de-repressed genes. This was closely in line with the findings by Margueron and colleagues that transcription and histone modification changes as a consequence of EZH2 loss are

predominantly irreversible (*Wassef et al., 2015*). Despite these irreversible changes to the epigenetic landscape, remarkably AAV-EED and AAV-HDAC1/2 were able to successfully remodel the chromatin landscape to functionally correct abnormalities of gene expression and organ function, even without correcting the genome-wide distribution of modified histone marks. These results reveal that it is possible to rehabilitate deregulated gene programs due to distorted chromatin landscapes, which contribute to diseases such as heart failure and congenital heart disease.

## Materials and methods

### Mice

All animal experiments were performed according to protocols (protocol #, Lsc-HeAB-1) approved by the Institutional Animal Care and Use Committees of Peking University and Boston Children's Hospital. *Eed*$^{fl}$ (*Xie et al., 2014a*) and *Myh6*$^{Cre}$ (*Agah et al., 1997*) alleles were described previously. 'WT' denotes either *Eed*$^{fl/fl}$; *Myh6*$^{Cre-}$ or *Eed*$^{fl/+}$ genotypes. Mice were injected with AAV (1 × 10$^{11}$ viral particles/gram body weight) by intraperitoneal or intravascular injection into pups and adults, respectively. Mice were intraperitoneally injected with SAHA (Sigma, 25 mg/kg/day) for the period of time as indicated. Echocardiography was conducted in either conscious or lightly anesthetized mice (isoflurane) using a Vevo 2100 imaging system (VisualSonics, Inc). Adult cardiomyocytes were isolated using type II collagenase in the Langendorff retrograde perfusion mode (*O'Connell et al., 2007*), and cardiomyocytes purity was evaluated for by co-immunostaining for cardiomyocyte mark Troponin i 3 (TNNI3) and DAPI. Isolated cardiomyocytes will be used only if more than 95% cells are positive for TNNI3.

### Cell culture

HL-1 cardiomyocyte-like cells (*Claycomb et al., 1998*) were cultured and validated as described previously (*He et al., 2011*), and determined as free of mycoplasma contamination. Fully confluent HL-1 cells were transfected using RNAiMax (Life Technoligies) with a pool of TriFECTa Dicer-Substrate siRNAs (DsiRNAs) (Integrated DNA Technologies) against Eed (conserved in both mouse and rat). Scrambled DsiRNA was used as the control. Sequences are provided (*Supplementary file 2A*). Cells were analyzed 72 hr after transfection. Transfection efficiency was verified to be more than 90% using fluorescently labeled control duplex as the indicator.

### Western blot

Cells were lysed in Nuclear Lysis Buffer (NLB, 50 mM Tris-Hcl pH8.0, 150 mM NaCl, 1% Nonidet P-40 (NP40), 1 mM EDTA, and fresh 1 mM PMSF and protease inhibitor cocktail (PIC)) for 30 mins, and 1% SDS was added before homogenization on QIAshredder columns. Equal amounts of proteins were resolved on 7.5% or 10% SDS-polyacrylamide gels and immunoblotted with primary antibodies listed (*Supplementary file 2B*). EED antibody was a kind gift from D. Reinberg (*Margueron et al., 2008*).

### Protein-protein interactions

Co-immunoprecipitation was performed as described (*He et al., 2012b*) with minor modifications. Nuclei were isolated using Hypotonic Lysis Buffer (20 mM HEPES pH 7.5, 10 mM KCl, 1 mM EDTA, 0.1 mM Na3VO4, 0.1 mM 0.2% (vol/vol) NP-40, 10% (vol/vol) glycerol plus PIC) and then resuspended in lysis buffer (50 mM Tris-HCl pH 8.0, 150 mM NaCl, 0.5% NP-40, 1 mM EDTA, and fresh 1 mM PMSF and PIC). Lysates were treated with 20 U of Benzonase nuclease (Stratagene) for 1 hr at 4°C. After centrifugation at 16,000 x g for 20 min at 4°C, soluble materials were kept as nuclear extracts. Nuclear extracts were precleared with Protein G beads before incubation with 5 μg of the indicated antibodies prebound to Protein G beads or Anti-Flag Affinity gel (Sigma, A2220) or Anti-HA High-Affinity Matrix (Roche, 12013819001) for 4 hr or overnight at 4°C. After washing three times with lysis buffer, precipitated proteins were recovered by elution in the sample buffer.

### In vitro HDAC activity assay

In vitro deacetylation assay was performed using a colorimetric immunoassay for deacetylated histones (Epigentek, P-4034–96). Fifty nanogram recombinant HDAC2 (Active Motif 31343) and 0–100 ng

recombinant EED (Cayman Chemical 10628) or bovine serum albumin were incubated at 37°C for 1 hr, and determined from the absorbance at 450 nm.

## ChIP and ChIP-seq

Antibody-based ChIP was performed as described (*He and Pu, 2010*) using antibodies listed (*Supplementary file 2B*). EED ChIP-seq was performed in adult cardiomyocytes using bioChIP method as described previously (*He et al., 2012b*). Briefly, BirA mice were injected with AAV expressing the cardiomyocyte-specific promoter cTNT driving EED in fusion with FLAG and bio (fbio) epitope tags, where bio is 23 amino-acid sequence specifically biotinylated by the enzyme BirA, and cardiomyocytes were harvested for bioChIP (ChIP pulldown through streptavidin beads) as indicated. ChIP-qPCR values were expressed as fold-enrichment.

Illumina ChIP-seq libraries were prepared as described (*He et al., 2014*) from purified cardiomyocytes, sequenced on an Illumina Hiseq 2500, and analyzed as described (*He et al., 2014*). Briefly, filtered reads were aligned to the July 2007 assembly of the mouse genome (NCBI 37, mm9) using the Burrows-Wheeler Aligner (BWA) (*Li and Durbin, 2009*) with default settings. Reads with no more than 4% (2 bp) mismatches and uniquely mapped to reference genome were kept for downstream analyses. The sequencing data are summarized here (*Supplementary file 3*).

Peaks were identified using MACS2 (*Zhang et al., 2008*) (version 2.1.0), with the parameter settings (–keep-dup=1; –broad). Peaks were assigned to the gene with the closest TSS within 100 kb. Proximal was defined within ±5 kb of a gene's TSS. Remaining regions were defined as distal. Cardiac enhancers were identified by H3K27ac peaks using adult heart H3K27ac ChIP-seq data (*He et al., 2014*). BigWig files normalized per 10 million aligned reads were viewed using the Integrative Genomics Viewer (IGV) (version 2.3.59) (*Robinson et al., 2011*).

ChIP and input reads were normalized to 10 million total aligned reads for H3K27ac, EED, and HDAC2. To reduce background signals for H3K27me3 in Eed^CKO vs WT, we firstly implemented a normalizing factor quantified through ChIP experiments (the relative levels of ChIP derived DNA) to the reads intensities in Eed^CKO and WT (*Ferrari et al., 2014*). To further exclude the bias due to normalization, we further chose the second normalization manner, yielding the similar results (*Kranz et al., 2013*). Simply, H3K27me3 ChIP signals were standardized using z-score transformation of ChIP reads by calculating mean and standard deviation of each dataset without all peak regions, presumably called background vs input. DANPOS (version 2.2.2) (*Chen et al., 2013*) was used to calculate H3K27me3 and H3K27ac signals (ChIP reads minus input reads, kept only if the value >0) for aggregate plots and box plots. H3K27ac and H3K27me3 signals were calculated at 20 bp intervals and plotted using R (version 3.2.2) and ggplot2 (version 1.0.1). Genome-wide distributions of H3K27me3 and H3K27ac signals were determined by counting ChIP signals within non-overlapping 1 kb wide windows tiled over the mouse genome. The box plots were generated using the average ChIP signals of ±5 kb of genes' TSS.

## RNA expression

RNA was prepared from purified adult cardiomyocytes as described (*He et al., 2012a*). mRNA sequencing library was prepared with Script-seq v2 (Illumina), and sequenced on Illumina Hi-seq 2000 (PE100). The resulting sequences were mapped to the mouse genome mm9 with STAR (*Dobin et al., 2013*). Transcripts per million reads (TPM) and Fragments per kilobase of exon per million fragments (FPKM) were generated for further quantification by RSEM and Bowtie 2 (*Langmead and Salzberg, 2012*; *Li and Dewey, 2011*), and differentially expressed genes were called with edgeR (v3.14.0) with the following criteria: adjusted p-value<0.05 and fold change (FC) >1.5 or <0.67. The gene ontology (GO) enrichment analysis of DEGs was performed using Metascape (*Tripathi et al., 2015*). The top six GO terms with p-value<0.001 in the 'biological process' category were used.

## Quantitative PCR

Real-time PCR was used to measure relative ChIP enrichment or gene expression. Quantitative PCR was performed using Power Sybr Green Master Mix (Life Technologies). Primer sequences are listed here (*Supplementary file 2C*). ChIP-qPCR values were expressed as fold-enrichment.

## Statistical analysis

Student's two-tailed *t*-test was used to determine the significance of differences between two groups in all qPCR assays. For bar charts, data are presented as mean ± SEM. For violin plots, the center line indicates the median. Wilcoxon-Mann-Whitney test was used to compare aggregate curves and violin-bar plots. Statistical significance was indicated with: *p<0.05; **p<0.01; ***p<0.001.

## Accession codes

The high-throughput data used in this study are available through the Cardiovascular Development Consortium Server at https://b2b.hci.utah.edu/gnomex/. Sign in as guest and go to experiment #410R. The data have also been deposited at GEO (accession number GSE73771).

## Acknowledgements

We thank H. Cheng from Institute of Molecular Medicine, Peking-Tsinghua Center for Life Sciences, Peking University, B Zhou from the Institute of Nutritional Sciences and Shanghai Institute for Biological Sciences and Y Zhang from the National Laboratory of Plant Molecular Genetics, Shanghai Institute of Plant Physiology and Ecology, Shanghai Institutes for Biological Sciences, Chinese Academy of Sciences for valuable suggestions. We thank T Luo from College of Chemistry and Molecular Engineering, Peking University for providing critical reagents. AH was supported by a startup fund from Peking-Tsinghua Center for Life Sciences, Peking University, the 1000 Youth Talents Program of China, the National Natural Science Foundation of China (31571487), and a Scientist Development Grant from American Heart Association (13SDG14320001). WTP was supported by funding from the National Heart Lung and Blood Institute (U01HL098166 and HL095712), by an Established Investigator Award from the American Heart Association, and by charitable donations from Karen Carpenter, Edward Marram, and Gail Federici Smith. The content is solely the responsibility of the authors and does not necessarily represent the official views of the funding agencies.

## Additional information

### Funding

| Funder | Grant reference number | Author |
| --- | --- | --- |
| National Natural Science Foundation of China | 31571487 | Aibin He |
| National Institutes of Health | U01HL098166 | William T Pu |
| National Institutes of Health | HL095712 | William T Pu |

The funders had no role in study design, data collection and interpretation, or the decision to submit the work for publication.

### Author contributions

SA, Data curation, Formal analysis, Investigation, Methodology; YP, Formal analysis, Validation, Investigation, Visualization, Methodology, Writing—original draft, Writing—review and editing; CL, Data curation, Investigation, Methodology; FG, Methodology, Writing—review and editing; XY, Data curation, Validation, Methodology; YY, Validation, Investigation; QM, JC, ZL, TWP, WZ, YL, Investigation, Methodology; PZ, D-ZW, Methodology; HX, JY, Resources, Methodology; SHO, Resources; WTP, Writing—review and editing; AH, Conceptualization, Data curation, Formal analysis, Supervision, Funding acquisition, Investigation, Writing—original draft, Writing—review and editing

### Author ORCIDs

William T Pu, http://orcid.org/0000-0002-4551-8079
Aibin He, http://orcid.org/0000-0002-3489-2305

## Ethics

Animal experimentation: All animal experiments were performed according to protocols (protocol number: Lsc-HeAB-1) approved by the Institutional Animal Care and Use Committees of Peking University

## Additional files

### Supplementary files

• Supplementary file 1. Tables for differential peaks for H3K27me3 and H3K27ac, and differentially expressed genes in WT and CKO.

• Supplementary file 2. The list of siRNAs, antibodies information and PCR primers. (A) The sequence of the TriFECTa Dicer-Substrate siRNAs (DsiRNAs). (B) Antibodies information. (C) The list of PCR primers.

• Supplementary file 3. The list of next-generation sequencing data.

### Major datasets

The following dataset was generated:

| Author(s) | Year | Dataset title | Dataset URL | Database, license, and accessibility information |
|---|---|---|---|---|
| Ai S, Peng Y, He A | 2016 | Transcriptional repression by non-canonical EED stimulation of histone deacetylase activity is required for heart maturation independently of H3K27me3 | https://www.ncbi.nlm.nih.gov/geo/query/acc.cgi?acc=GSE73771 | Publicly available at the NCBI Gene Expression Omnibus (accession no: GSE73771) |

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
