## [Decision Letter]

Thank you for submitting your article "EED orchestration of heart maturation through interaction with HDACs is H3K27me3-independent" for consideration by *eLife*. Your article has been favorably evaluated by Kevin Struhl (Senior Editor) and three reviewers, one of whom is a member of our Board of Reviewing Editors. The following individuals involved in review of your submission have agreed to reveal their identity: Mauro Calabrese (Reviewer #3).

The reviewers have discussed the reviews with one another and the Reviewing Editor has drafted this decision to help you prepare a revised submission.

First: There is general agreement that your manuscript is a potentially important study and that the findings you report are surprising. There is enthusiasm for publishing this paper in *eLife*. However, the surprising and significant conclusions rest heavily on there being no cardiac cell division after birth to dilute away the H3K27 mark. Recent papers suggest that heart cells can and do divide after birth. The authors must test this to see if cell division is taking place. Current models are that non-dividing cells should be able to maintain H3K27me3 without PRC2, which is why the paper concludes that PRC2 must have another function independent of H3K27me3 in light of the cardiomyopathy seen after EED deletion.

Second: The reviewers also commented that the Text understated the loss of the histone mark. One third of derepressed genes did show loss of H3K27me3. How did this take place? This could have been due to cell division and passive loss of the histone mark. Quantitative measurements with statistical testing should be performed for known polycomb targets in heart muscle.

Third: there is general concern that the EED KO may not be a true null. Peptides can still be seen in Western. It would be important to determine whether the mutant protein truly has no activity.

Finally: your paper proposes a direct mechanistic link between EED and HDAC. One reviewer suggested the excellent idea of treating cardiomyocytes with HDAC-inhibitors to see if the same target genes are upregulated as in the EED CKO. If so, the results would speak for a common mechanism.

For publication, we would like to see these points and related points addressed in a revised manuscript. If you choose to revise your manuscript, we would also require a reviewer-link to the GEO data for this study. We hope to see a revised manuscript within 2-3 months, although the nature of the requested experiments may require you to take more time.

*Reviewer #1:*

This is a potentially important study that proposes that depleting EED (and therefore PRC2) in postnatal cardiomyocytes results in a lethal dilated cardiomyopathy with loss of silencing of myocardial gene program. This finding is surprising because non-dividing cells are thought to be able to maintain H3K27me3 without PRC2. The authors conclude that EED has a non-canonical function that is independent of its H3K27 methylation activity. The paper reports exciting findings. However, there are two concerns that must be addressed for publication.

1) Postnatal cardiomyocytes have been shown in recent studies to proliferate. Please consult (as one example): Naqvi N, Li M, Calvert JW, Tejada T, Lambert JP, Wu J, Kesteven SH, Holman SR, Matsuda T, Lovelock JD, et al. 2014. A proliferative burst during preadolescence establishes the final cardiomyocyte number. Cell 157: 795-807. The authors' findings relating to an EED deletion could be explained if there is proliferation in the postnatal cardiomyocytes. In this case, there would not be a "non-canonical" function of PRC2, and the authors would have to tone down their conclusions considerably. The paper may still be publishable without this dramatic conclusion.

2) The authors' data suggest that there is in fact some loss of H3K27me3. This would be consistent with continued (though limited) proliferation of cardiomyocytes after birth. I think the authors underestimated the depletion of the H3K27me3 mark.

*Reviewer #2:*

In this manuscript, the authors describe the interesting observation that EED is critical for heart maturation and propose a novel mechanism, in which EED does not repress its target genes via its chanonical role in PRC2 through applying the H3K27me3 mark, but rather through affecting histone acetylation by modulating HDAC activity. The authors present clear evidence for a striking phenotype, and provide a comprehensive analysis with convincing rescue experiments showing a clear role for EED and HDACs. However, some discrepancies in the data, like for example opposing results for recruitment of HDACs by EED depending on the assay (EED loss of function – no effect vs. sufficiency via EED tethering), need to be addressed before firm mechanistic claims linking EED with HDAC activity can be made.

1) Figure 1 and Figure 1—figure supplement 1: There is still a remaining Eed transcript detectable in Eed^CKO^ mice. The Western Blot is cut, so it is not visible, if there is a remaining Eed protein of reduced size. The authors should show the entire Western Blot and quantify the remaining RNA-levels based on qPCR (both for wt and CKO transcript levels). It is possible that a truncated Eed protein could still have a remaining function in the CKO allele, or incomplete deletion by the Cre-driver (some remaining protein visible in Figure 1) could explain the leftover H3K27me3 signal. A truncated EED-protein could also have different hypomorphic functionalities, for example related to gene repression, than a complete null would have. Therefore, it is important to clarify, if the presented CKO allele is a complete functional null or not.

2) Figure 4: In Figure 4 the authors have demonstrated that EED does not play a significant role in vivo in recruiting HDAC2 to chromatin. However, artificial tethering of EED to ectopic sites led to strong HDAC2 recruitment by dCas9-EED (Figure 4). Could this be evidence for a remaining EED-protein in the CKO allele that might still able to recruit HDAC2 but not to stimulate its activity (related to my concern mentioned in point 1)? It would be interesting to know, if a complete Eed KO shows similar effects on HDAC recruitment/activity. Could this be tested in a full EED KO or knockdown cell line?

3) Figure 5: While it is interesting that overexpression of HDAC1/2 leads to a partial rescue (changes in H3K27ac look not very strong after HDAC overexpression) of the EED CKO phenotype, a complementary proof would be a loss of function experiment for HDAC activity in heart muscle cells. It would make the argument more convincing, if treatment of wt heart muscle cells with HDAC-inhibitors could show, if the same genes, which are upregulated in EED CKO, also become upregulated after HDAC inhibition. This would strengthen the link between EED and HDAC activity more directly and make the proposed model more convincing.

*Reviewer #3:*

This is an impressive study that demonstrates the functions and mechanisms of Eed-mediated gene regulation in the developing heart. The authors find that deletion of Eed during heart development results in cardiomyopathy concomitant with upregulation of several slow-twitch myofiber genes. A battery of genomic profiling experiments performed in wild-type, mutant, and Eed or HDAC rescued cardiomyocytes, demonstrate that Eed recruits and/or stimulates histone deacetylase activity at key target genes to acutely repress their expression during heart maturation. The major surprise is that this repressive effect is caused by recruitment and/or stimulation of HDAC activity, and is independent of major changes in H3K27me3 at two-thirds of target genes. The experiments presented are thorough and extensive. The results corroborate prior works that have linked PRC2 complex to HDACs, and should be of broad interest to many researchers. I think the manuscript could almost be published as is.

1) From a mechanistic standpoint, the incomplete loss of H3K27me3 upon EED knockout left me scratching my head, because loss of Eed causes a global loss of H3K27me3 in just about every other cell type in which an Eed deletion has been performed. The authors suggest that the lack of complete H3K27me3 loss upon Eed deletion is due to the fact that the modification is very stable in cells that do not divide, like cardiomyocytes.

An alternate explanation for the persistence of H3K27me3 upon Eed deletion is that the knockout is incomplete and/or there remains some functional PRC2 in the cells. For example, the Cre-driver may not KO Eed in 100% of cells. Or (less likely), maybe Ezh2 can be recruited to some sites independently of Eed.

The authors should do a few simple characterization experiments in WT vs. CKO cardiomyocytes to clear up the uncertainty. A PCR-based assay to detect the floxed vs. deleted allele, and an immunostain to Eed would be informative. Also, Ezh2 is known to be destabilized in the absence of Eed. Thus, a western blot and immunostain to Ezh2 would also be informative.

Also, does the Eed deletion generate a premature truncation of the protein? Or is the deletion in frame?

2) Along the lines of 1, the authors should better explain the Eed western blot in Figure 1. Why are there three bands for Eed? The largest band does not disappear upon EED CKO (it is marked by an * in the figure). Could this be residual Eed? Can the authors use another antibody to rule out if it is a non-specific band, and not an alternate Eed isoform?

---

## [Author Response]

*First: There is general agreement that your manuscript is a potentially important study and that the findings you report are surprising. There is enthusiasm for publishing this paper in eLife. However, the surprising and significant conclusions rest heavily on there being no cardiac cell division after birth to dilute away the H3K27 mark. Recent papers suggest that heart cells can and do divide after birth. The authors must test this to see if cell division is taking place. Current models are that non-dividing cells should be able to maintain H3K27me3 without PRC2, which is why the paper concludes that PRC2 must have another function independent of H3K27me3 in light of the cardiomyopathy seen after EED deletion.*

Our main finding is that upregulation of the majority of genes in postnatal cardiomyocyte EED knockout was not linked to loss of H3K27me3. This observation is independent of whether cardiomyocyte proliferate or not. However, the paucity of postnatal cardiomyocyte proliferation is likely an important reason that our data differs from studies of EED in cell lines or proliferating cell types.

Although Naqvi et al. (PMID:24813607) suggested a burst of cardiomyocyte proliferation in mouse preadolescence (2 weeks after birth), this finding has not been reproduced by the work of four independent groups using multiple independent methods (PMID: 24876275, 26544945, 26544927, and 26472817). The general consensus currently is that there is detectable but limited cardiomyocyte proliferation in the postnatal heart (for example, PMID 17660827, 19342590, 23222518, in addition to the 4 studies cited in the previous sentence). The large majority of murine cardiomyocytes undergo a final round of nuclear division within the first postnatal week of life, and then terminally exit the cell cycle. In the mature adult heart, there is general agreement that the rate of cardiomyocyte turnover is low (CM turnover of 0.7-1% per year in adult mouse and human). We performed immunofluorescent staining for phosphohistone H3 (pH3, M-phase marker) of heart sections at three different postnatal ages and confirmed that there is little cardiomyocyte cell cycle activity after the first postnatal week (Figure 8). Given the large body of data on this topic in the literature and the elaborate methods (C14 carbon dating; imaging mass spectrometry; genetic pulse-labeling) that have been employed to reach the current consensus, we felt that including these data in this paper would not add to the paper or to the overall literature on this subject.

Author response image 1.Examination of cariomyocyte proliferation rate in neonatal heart.Immunostaining and quantification of phosphorylated histone H3 (pH2; red) as a marker for M-phase in hearts at P0, P5 and P18 cardiomyocytes (TNN13, green). Scale bar = 20 μm.**DOI:**
http://dx.doi.org/10.7554/eLife.24570.019

*Second: The reviewers also commented that the Text understated the loss of the histone mark. One third of derepressed genes did show loss of H3K27me3. How did this take place? This could have been due to cell division and passive loss of the histone mark. Quantitative measurements with statistical testing should be performed for known polycomb targets in heart muscle.*

As noted by the reviewers, 1/3 of upregulated genes had H3K27me3 loss, whereas 2/3 of upregulated genes had retained H3K27me3. This variation in H3K27me3 levels strongly argues that loss of H3K27me3 is not due to passive dilution during proliferation. We have done separate experiments that measure locus-specific histone turnover in cardiomyocytes, which is possible because of their low turnover rate. We observed that cardiomyocytes display finely-regulated locus-specific histone turnover. This separate manuscript is currently in preparation.

In the revised manuscript, we have made more clear that 1/3 of genes are upregulated and lose H3K27me3, consistent with the canonical model of PRC2-mediated repression through H3K27me3 deposition. We grouped the upregulated genes into clusters based on their H3K27me3 signal in WT and Eed^CKO^ (Figure 1). C1 corresponds to the cluster with gene upregulation and H3K27me3 loss. The majority of genes fall into C2 and C3, which do not show this canonical association. As shown in Figure 1, we quantitatively compared H3K27me3 signal between WT and Eed^CKO^ within each cluster with appropriate statistical testing to better demonstrate the behavior of H3K27me3 in C1 vs C2 and C3.

In the revision, we also performed quantitative analysis of H3K27me3 levels on known polycomb targets (defined by ± 10 kb of gene TSS) in cardiomyocytes as suggested by the editor and reviewers (Figure 1—figure supplement 1). We found that median H3K27me3 levels were reduced only in EED target genes among the quartile of upregulated genes with the strongest H3K27me3 in WT (Figure 1—figure supplement 2), again indicating that H3K27me3 loss was only seen in a minority of upregulated genes.

*Third: there is general concern that the EED KO may not be a true null. Peptides can still be seen in Western. It would be important to determine whether the mutant protein truly has no activity.*

In Figure 1, the above 75 kDa, indicated by asterisk, is larger than the calculated full-length molecular weight of 60 kDa and is a non-specific band. The remaining EED detected in EED-CKO is likely due to the signals from non-cardiomyocytes of P0 and P5 heart apex used for Western blot and RT-qPCR (Figure 1—figure supplement 1). While Figure 1 used whole heart apex, the rest of the manuscript used isolated adult cardiomyoctes with >95% purity for Western blots, RNA analysis, and ChIP-seq.

In the EED conditional allele, the floxed exons are 3-6, which contain the critical WD-repeat domain of EED that binds to nucleosomes and EZH2. This same floxed allele was tested by Stuart Orkin’s lab and demonstrated to be functionally null (Xie, et al., Cell Stem Cell, 2014, PMID: 24239285). In RNA-seq of purified cardiomyocytes, exons 3-6 were not detectably expressed, confirming high efficiency gene ablation (Figure 1—figure supplement 2). We further verified the nearly complete excision of the floxed allele by PCR analysis with DNA from purified cardiomyocytes (Figure 1—figure supplement 2). Based on these data, we are confident that there is no residual, functional EED protein in the large majority of adult cardiomyocytes used for the RNA-seq and ChIP-seq experiments.

*Finally: your paper proposes a direct mechanistic link between EED and HDAC. One reviewer suggested the excellent idea of treating cardiomyocytes with HDAC-inhibitors to see if the same target genes are upregulated as in the EED CKO. If so, the results would speak for a common mechanism.*

Our model is that EED stimulates HDAC activity, and therefore HDAC inhibition might mimic some of the features of EED-CKO. As suggested by the editor and reviewers, we treated control (Het + AAV-luc) mice with the broad spectrum HDAC inhibitor SAHA (Figure 6). SAHA did not significantly affect heart function or derepress genes that were upregulated in EED-CKO. These data indicate that EED-HDAC is not required in normal cardiomyocyte maturation, possibly due to multiple, redundant gene repressive mechanisms. However, when EED-CKO mice were rescued with AAV9-EED, rescue was blocked by SAHA (Figure 6). This suggests that transient EED loss of function disrupts the chromatin landscape and multiple gene repressive mechanisms. Whereas re-expression of EED can restore gene repression, this process depends on HDAC activity.

*For publication, we would like to see these points and related points addressed in a revised manuscript. If you choose to revise your manuscript, we would also require a reviewer-link to the GEO data for this study. We hope to see a revised manuscript within 2-3 months, although the nature of the requested experiments may require you to take more time.*

We have deposited the data to GEO and have supplied a reviewer link in the revised manuscript.

*Reviewer #1:*

*This is a potentially important study that proposes that depleting EED (and therefore PRC2) in postnatal cardiomyocytes results in a lethal dilated cardiomyopathy with loss of silencing of myocardial gene program. This finding is surprising because non-dividing cells are thought to be able to maintain H3K27me3 without PRC2. The authors conclude that EED has a non-canonical function that is independent of its H3K27 methylation activity. The paper reports exciting findings. However, there are two concerns that must be addressed for publication.*

*1) Postnatal cardiomyocytes have been shown in recent studies to proliferate. Please consult (as one example): Naqvi N, Li M, Calvert JW, Tejada T, Lambert JP, Wu J, Kesteven SH, Holman SR, Matsuda T, Lovelock JD, et al. 2014. A proliferative burst during preadolescence establishes the final cardiomyocyte number. Cell 157: 795-807. The authors' findings relating to an EED deletion could be explained if there is proliferation in the postnatal cardiomyocytes. In this case, there would not be a "non-canonical" function of PRC2, and the authors would have to tone down their conclusions considerably. The paper may still be publishable without this dramatic conclusion.*

We addressed this comment in detail in the summary critique (point #1). The main observation that gene upregulation was not associated with H3K27me3 loss at 2/3 of genes is not dependent on whether or not cardiomyocytes proliferate. The terminal differentiation of postnatal CMs is relevant to the study because it provides a potential explanation for why this non-canonical mechanism is not observed in intensively studied cell types such as ESCs and cancer cell lines, where cell proliferation demands PRC2 deposition of new H3K27me3 and hence emphasizes this aspect of PRC2 function.

Although Naqvi et al. (PMID:24813607) showed a burst of cardiomyocyte proliferation in mouse preadolescence (2 weeks after birth), this result has not been reproduced by four independent groups using multiple independent approaches (PMID: 24876275, 26544945, 26544927, and 26472817). The general consensus currently is that there is detectable but limited cardiomyocyte proliferation in the postnatal heart. The large majority of murine cardiomyocytes undergo a final round of nuclear division within the first postnatal week of life, and then terminally exit the cell cycle. In the mature adult heart, there is general agreement that the rate of cardiomyocyte turnover is low (CM turnover of 0.7-1% per year in adult mouse and human). Figure 8 shows additional data that cardiomyocytes have largely exited the cell cycle after the first postnatal week. We do not rule out all proliferation-related dilution of H3K27me3 in EED-CKO, but it is likely to make a small contribution to the loss of H3K27me3.

We observed that H3K27me3 is retained at some sites and lost at others. This cannot be accounted for simply by proliferation-related dilution, and further indicates that other mechanisms are responsible for H3K27me3 loss. We have studied histone dynamics in postnatal cardiomyocytes extensively and are preparing a separate manuscript on this topic. However, it is outside of the scope of the present study.

*2) The authors' data suggest that there is in fact some loss of H3K27me3. This would be consistent with continued (though limited) proliferation of cardiomyocytes after birth. I think the authors underestimated the depletion of the H3K27me3 mark.*

We agree that about 1/3 of derepressed genes have reduced H3K27me3. On the other hand, 2/3 of these genes retain H3K27me3, which was quantified in Figure 1 and also in this revision (Figure 1—figure supplement 2). This differential retention of H3K27me3 cannot be accounted for by passive dilution during proliferation and strongly argues for other mechanisms of H3K27me3 loss. We have done separate experiments that measure locus-specific histone turnover in cardiomyocytes, which is possible because of their low turnover rate. We observed that cardiomyocytes display finely-regulated locus-specific histone turnover. This separate manuscript is currently in preparation.

The major focus of this manuscript is understanding the mechanism of derepression of the 2/3 of genes that retained H3K27me3.

*Reviewer #2: In this manuscript, the authors describe the interesting observation that EED is critical for heart maturation and propose a novel mechanism, in which EED does not repress its target genes via its chanonical role in PRC2 through applying the H3K27me3 mark, but rather through affecting histone acetylation by modulating HDAC activity. The authors present clear evidence for a striking phenotype, and provide a comprehensive analysis with convincing rescue experiments showing a clear role for EED and HDACs. However, some discrepancies in the data, like for example opposing results for recruitment of HDACs by EED depending on the assay (EED loss of function – no effect vs. sufficiency via EED tethering), need to be addressed before firm mechanistic claims linking EED with HDAC activity can be made. 1) Figure 1 and Figure 1—figure supplement 1: There is still a remaining Eed transcript detectable in EedCKO mice. The Western Blot is cut, so it is not visible, if there is a remaining Eed protein of reduced size. The authors should show the entire Western Blot and quantify the remaining RNA-levels based on qPCR (both for wt and CKO transcript levels). It is possible that a truncated Eed protein could still have a remaining function in the CKO allele, or incomplete deletion by the Cre-driver (some remaining protein visible in Figure 1) could explain the leftover H3K27me3 signal. A truncated EED-protein could also have different hypomorphic functionalities, for example related to gene repression, than a complete null would have. Therefore, it is important to clarify, if the presented CKO allele is a complete functional null or not.*

In Figure 1, the band at 75 kDa, indicated by an asterisk, is larger than the expected full-length molecular weight of 60 kDa and represents a non-specific signal. The remaining EED detected in EED CKO is likely due to the signals from non-cardiomyocytes of P0 and P5 heart apex used for Western blot and RT-qPCR (Figure 1—figure supplement 1). While Figure 1 used whole heart apex, the rest of the manuscript used isolated adult cardiomyoctes with >95% purity for the other Western blots, RNA analysis, and ChIP-seq. In the EED conditional allele, the floxed exons are 3-6, which contain the critical WD-repeat domain of EED that binds to nucleosomes and EZH2. This same floxed allele was tested by Stuart Orkin’s lab and demonstrated to be functionally null (Xie, et al., Cell Stem Cell, 2014, PMID: 24239285). In RNA-seq of purified cardiomyocytes, exons 3-6 were not detectably expressed, confirming high efficiency gene ablation (Figure 1—figure supplement 2). We further verified the nearly complete excision of the floxed allele by PCR analysis of DNA from purified cardiomyocytes (Figure 1—figure supplement 2). Based on these data, we are confident that there is no residual, functional EED protein in the large majority of adult cardiomyocytes used for the RNA-seq and ChIP-seq experiments.

*2) Figure 4: In Figure 4 the authors have demonstrated that EED does not play a significant role in vivo in recruiting HDAC2 to chromatin. However, artificial tethering of EED to ectopic sites led to strong HDAC2 recruitment by dCas9-EED (Figure 4). Could this be evidence for a remaining EED-protein in the CKO allele that might still able to recruit HDAC2 but not to stimulate its activity (related to my concern mentioned in point 1)? It would be interesting to know, if a complete Eed KO shows similar effects on HDAC recruitment/activity. Could this be tested in a full EED KO or knockdown cell line?*

We thank the reviewer for this critical point. Although the tethering experiment demonstrated that EED can stimulate local histone deacetylation, it is confusing because it appears to contradict the in vivo data that show that EED is dispensable for recruiting HDAC2 to chromatin. The cellular and chromatin contexts of these experiments are quite different and we do not believe the results are incompatible. For instance, the tested genes do not naturally have HDAC2 or EED occupancy, whereas the in vivo experiment was a loci that do have endogenous HDAC2 and EED occupancy. Nevertheless, because this experiment is confusing, we removed it from the revised manuscript.

*3) Figure 5: While it is interesting that overexpression of HDAC1/2 leads to a partial rescue (changes in H3K27ac look not very strong after HDAC overexpression) of the EED CKO phenotype, a complementary proof would be a loss of function experiment for HDAC activity in heart muscle cells. It would make the argument more convincing, if treatment of wt heart muscle cells with HDAC-inhibitors could show, if the same genes, which are upregulated in EED CKO, also become upregulated after HDAC inhibition. This would strengthen the link between EED and HDAC activity more directly and make the proposed model more convincing.*

We thank the reviewer for this excellent suggestion. We treated control (Het + AAV-luc) mice with the broad spectrum HDAC inhibitor SAHA (Figure 6). SAHA did not significantly affect heart function or derepress genes that were upregulated in EED-CKO. These data indicate that EED-HDAC is not required in normal cardiomyocyte maturation, possibly due to multiple, redundant gene repressive mechanisms. However, when EED-CKO mice were rescued with AAV9-EED, rescue was blocked by SAHA (Figure 6). This suggests that transient EED loss of function disrupts the chromatin landscape and multiple gene repressive mechanisms. Whereas re-expression of EED can restore gene repression, this process depends on HDAC activity.

*Reviewer #3:*

*[…] 1) From a mechanistic standpoint, the incomplete loss of H3K27me3 upon EED knockout left me scratching my head, because loss of Eed causes a global loss of H3K27me3 in just about every other cell type in which an Eed deletion has been performed. The authors suggest that the lack of complete H3K27me3 loss upon Eed deletion is due to the fact that the modification is very stable in cells that do not divide, like cardiomyocytes.*

*An alternate explanation for the persistence of H3K27me3 upon Eed deletion is that the knockout is incomplete and/or there remains some functional PRC2 in the cells. For example, the Cre-driver may not KO Eed in 100% of cells. Or (less likely), maybe Ezh2 can be recruited to some sites independently of Eed.*

We demonstrated that *Ezh*2 is below detection level by Western blot in adult cardiomyocytes (He, et al. 2012, Genes & Dev.). We believe that H3K27me3 persists largely because of low histone turnover rates, i.e. H3K27me3 that was deposited prior to EED deletion persists for weeks in the absence of de novo deposition. This view is supported by a separate study in which we have studied histone turnover in terminally differentiated cardiomyocytes (in preparation). In the revised manuscript, we provide several additional lines of evidence to show that Cre-mediated EED recombination was highly efficient in CMs (Figure 1—figure supplement 1; Figure 1—figure supplement 2).

*The authors should do a few simple characterization experiments in WT vs. CKO cardiomyocytes to clear up the uncertainty. A PCR-based assay to detect the floxed vs. deleted allele, and an immunostain to Eed would be informative. Also, Ezh2 is known to be destabilized in the absence of Eed. Thus, a western blot and immunostain to Ezh2 would also be informative.*

In RNA-seq of purified cardiomyocytes, exons 3-6 were not detectably expressed, confirming high efficiency gene ablation (Figure 1—figure supplement 2). We further verified the nearly complete excision of the floxed allele by PCR analysis with DNA from purified cardiomyocytes (Figure 1—figure supplement 2). The immunostaining data showed loss of H3K27me3 signal in CMs (Figure 1—figure supplement 2). Unfortunately, there is no commercial antibody suitable for EED immunofluorescent staining that we are aware of. Based on these data, we are confident that there is no residual, functional EED protein in the large majority of adult cardiomyocytes used for the RNA-seq and ChIP-seq experiments. As Ezh2 expression is very low in wild-type adult cardiomyocytes at the transcript level (He et al. 2012 Circ Res.), the proposed experiment on EZH2 would not be informative.

*Also, does the Eed deletion generate a premature truncation of the protein? Or is the deletion in frame?*

The floxed exons are 3-6, which contain the critical WD-repeat domain of EED that binds to nucleosome and EZH2. This same floxed allele was tested by Stuart Orkin’s lab and demonstrated to be functionally null (Xie, et al., Cell Stem Cell, 2014, PMID: 24239285). Their data suggests that it does not generate a truncated protein but rather leads to loss of protein.

*2) Along the lines of 1, the authors should better explain the Eed western blot in Figure 1. Why are there three bands for Eed? The largest band does not disappear upon EED CKO (it is marked by an * in the figure). Could this be residual Eed? Can the authors use another antibody to rule out if it is a non-specific band, and not an alternate Eed isoform?*

The largest band marked by "*" is a non-specific band. We made this more clear in the figure legend. The size of this band is larger than the size expected for full length EED. There are weak bands of remaining EED in the EED CKO that are likely due to signals from non-cardiomyocytes of P0 and P5 heart apex used for Western blot and RT-qPCR (Figure 1—figure supplement 1). PCR from genomic DNA of purified CMs (Figure 1—figure supplement 2) confirmed nearly complete excision of floxed exons, and this was independently supported by RNA-seq (Figure 1—figure supplement 2).

EED has multiple different isoforms: the Reinberg group, the Magnuson group, and our group showed that four EED isoforms exist and each varies during ESCs differentiation and heart development (Kuzmichev, 2005, PNAS; Montgomery, 2007, J. Mol. Biol.; He, 2012, Circ. Res.). These isoforms account for the multiple bands of EED on the western blot.